# Does maximization of net carbon profit enable the prediction of vegetation behaviour in savanna sites along a precipitation gradient?

Remko C. Nijzink[1], Jason Beringer[2], Lindsay B. Hutley[3], and Stanislaus J. Schymanski[1]

[1]Luxembourg Institute of Science and Technology, Environmental Research and Innovation, Catchment and Eco-hydrology Research Group, Belvaux, Luxembourg
[2]School of Agriculture and Environment, The University of Western Australia, Crawley, WA, Australia, 6909
[3]Research Institute for the Environment and Livelihoods, Charles Darwin University, Darwin, NT, Australia, 0909

**Correspondence:** R.C. Nijzink (remko.nijzink@list.lu)

**Abstract.**

Most terrestrial biosphere models (TBMs) rely on more or less detailed information about the properties of the local vegetation. In contrast, optimality-based models require much less information about the local vegetation as they are designed to predict vegetation properties based on general principles related to natural selection and physiological limits. Although such models are not expected to reproduce current vegetation behaviour as closely as models that use local information, they promise to predict the behaviour of natural vegetation under future conditions, including the effects of physiological plasticity and shifts of species composition, which are difficult to capture by extrapolation of past observations.

A previous model intercomparison using conventional TBMs revealed a range of deficiencies in reproducing water and carbon fluxes for savanna sites along a precipitation gradient of the North Australian Tropical Transect (Whitley et al., 2016). Here, we examine the ability of an optimality-based model (the Vegetation Optimality Model, VOM) to predict vegetation behaviour for the same savanna sites. The VOM optimizes key vegetation properties such as foliage cover, rooting depth and water use parameters in order to maximize the Net Carbon Profit (NCP), defined as the difference between total carbon taken up by photosynthesis minus the carbon invested in construction and maintenance of plant organs.

Despite a reduced need for input data, the VOM performed similarly or better than the conventional TBMs in terms of reproducing the seasonal amplitude and mean annual fluxes recorded by flux towers at the different sites. It had a relative error of 0.08 for the seasonal amplitude in ET, and was among the best three models tested with the smallest relative error in the seasonal amplitude of gross primary productivity (GPP). Nevertheless, the VOM displayed some persistent deviations from observations, especially for GPP, namely an underestimation of dry season evapo-transpiration at the wettest site, suggesting that the hydrological assumptions (free drainage) have a strong influence on the results. Furthermore, our study exposes a persistent overprediction of vegetation cover and carbon uptake during the wet seasons by the VOM. Our analysis revealed several areas for improvement in the VOM and the applied optimality theory, including a better representation of the hydrological settings, as well as the costs and benefits related to plant water transport and light capture by the canopy.

The results of this study imply that vegetation optimality is a promising approach to explain vegetation dynamics and the resulting fluxes. It provides a way to derive vegetation properties independently from observations, and allows for a more insightful evaluation of model shortcomings as no calibration or site-specific information is required.

# 1 Introduction

Current state-of-the-art terrestrial biosphere models (TBMs), i.e. land surface models, dynamic global vegetation models and stand-scale models (Whitley et al., 2016), commonly rely on locally observed vegetation properties and/or phenology and informed guesses where observations to calibrate parameters are not available (Whitley et al., 2016). At the same time, many model parameters are assumed to be invariant in time and space, such as prescribed rooting depth, which does not adapt to seasonal and longer-term changes in environmental conditions. As a result, TBMs often struggle to reproduce the dynamics of observed carbon and water fluxes, notably in seasonal environments such as savanna ecosystems, and are often outperformed even by simple regression models (Best et al., 2015; Whitley et al., 2016). Furthermore, calibration of vegetation properties on past observations may limit the utility of the models for predictions in a changing environment (Schulz et al., 2001) or outside the environment used to develop and/or parameterize them. Therefore, novel methods to capture the dynamic adaptation of vegetation-related parameters in TBMs are urgently required given changing climates and disturbance regimes (Schulz et al., 2001; Whitley et al., 2016).

At the same time, savanna ecosystems are extremely complex to model, but also contribute up to 30% of the global net primary productivity (Grace et al., 2006; Lehmann et al., 2014). Therefore, understanding their behaviour is crucial for predicting the effects of global change. At a global-scale, the distribution of savannas are tightly coupled to the occurrence of wet-dry seasonal climates, namely the savanna climate type as defined by Köppen-Geiger (Peel et al., 2007; Beck et al., 2018). These ecosystems are highly dynamic due to the seasonality of the climate which in turn drives dramatic changes in phenology and productivity of both woody and grassy components (Scholes and Archer, 1997; House et al., 2003). The combination of co-occurring seasonal grasses, relatively deep-rooting perennial trees, frequent fire and strong re-sprouting capacity in savannas presents a set of challenges for vegetation models, which struggle to reproduce variation in soil water and vegetation cover, particularly between overstorey and understorey vegetation (Baudena et al., 2015). For these reasons, the savanna sites along the North Australian Tropical Transect (NATT) provide an excellent living laboratory (Hutley et al., 2011), especially as the strong rainfall gradient from north to south (approx. a decrease of 1 mm mean annual rainfall per km south) provides different climatological circumstances, whereas other factors as topography and soils remain rather constant.

Using the savanna sites of the NATT, Whitley et al. (2016) was able to reveal a range of deficiencies in different TBMs . Most of the models required information about vegetation cover as an input, whereas only one of the models (LPJ-GUESS) represented vegetation in a dynamic way. This reflects also the wider spectrum of TBMs, as few models predict vegetation dynamics, and the few that do, rely on prescribed behaviour of a few plant functional types deemed representative for a given site. In addition, as Whitley et al. (2016) pointed out, the interplay between shallow rooted (0.5-1.0 m) seasonal vegetation and deep-rooted (>2.0 m) perennial vegetation is an important factor controlling fluxes in savannas. This relates to the strong seasonality in water availability in these ecosystems, which leads to issues for models that prescribe constant rooting depths of around 2.0 m or less. Finally, Whitley et al. (2016) showed the importance of disturbances, most notably fires, in shaping the community composition, structure and fluxes in savannas (Scheiter et al., 2013), but the models in their study did not consider such disturbances. So far, there are only a few models that explicitly model savanna-like fire events, such as aDGVM (Scheiter

and Higgins, 2009). This model has been used to model spatial and temporal patterns of biomass accumulation in both African (Scheiter and Higgins, 2009) and Australian savannas (Scheiter et al., 2015) and demonstrated how fire management practices and climate change may influence carbon uptake and storage in savanna vegetation.

More in general, model intercomparison studies have revealed several persistent deficiencies of TBMs, often related to the representation of vegetation and the latent heat flux (Pitman et al., 1999, 2009; Best et al., 2015), or the carbon cycle (Teckentrup et al., 2021). At the same time, only a small number of models simulate vegetation dynamics in a prognostic way, such as LPJ-GUESS (Smith et al., 2001), Tethys-Chloris (Fatichi et al., 2012) or RHESSys (Tague, 2004). TBMs that require site-specific vegetation properties, such as leaf area index or rooting depths, include widely used models such as the Soil, Plant, Atmosphere model (SPA, Williams et al., 1996a) and BESS (Ryu et al., 2011, 2012). These often produce satisfactory results, but they strongly depend on the quality of the prescribed vegetation data. Therefore, these models are of limited utility for modelling responses to changing environmental conditions, which is particularly important for longer-term predictions.

In order to improve the generality of TBMs, the implementation of organizing principles, such as optimality, has been proposed by an increasing number of scientists (e.g. Eagleson, 1982; Rodríguez-Iturbe and Rinaldo, 2001; McDonnell et al., 2007; Schymanski et al., 2007; Franklin et al., 2012; Bonan et al., 2014; De Kauwe et al., 2015; Haverd et al., 2016; Buckley et al., 2017; Wang et al., 2017, 2018; Franklin et al., 2020). Optimality assumes that a system self-optimizes in order to maximize or minimize some goal function related to plant survival (e.g. maximum productivity or minimum stress) or self-maintenance of system compartments, such as entropy production or power extraction (Schymanski et al., 2009a). This means that system properties and behaviour with trade-offs related to said goal function can be predicted rather than prescribed based on past observations or calibration.

Following this paradigm, Schymanski et al. (2007, 2009b) proposed that vegetation may self-optimize to maximize its Net Carbon Profit (NCP), defined as the total difference in carbon assimilated by photosynthesis and the carbon costs for the construction and maintenance of the plant tissues. These principles were implemented in the Vegetation Optimality Model (VOM, Schymanski et al., 2009b, 2015) that explicitly models vegetation dynamics by optimizing vegetation properties to maximize the NCP. The VOM has been applied successfully by Schymanski et al. (2009b) and Schymanski et al. (2015), but several challenges can still be identified.

So far, the number of studies that test these optimality principles remains limited, and showed also varying levels of success. Schymanski et al. (2015) applied this principle to simulate effects of elevated atmospheric $CO_2$-concentrations on vegetation across several sites in Australia and found that the results were consistent with experimental evidence. In a similar study, Wang et al. (2018) found that the inclusion of the maximum NCP optimality principle to model optimal root systems improved the simulation of water use by phreatophytic vegetation in the widely used land surface model Noah-MP. In contrast to these successful applications of optimality, Dekker et al. (2010) reported a 38% overestimation of $CO_2$-assimilation, when applying the maximum NCP principle in a Douglas fir plantation in the Netherlands. It is not clear if this bias was due to invalidity of the maximum NCP principle, missing model constraints, the short duration over which the optimization was performed (1 year) or the limitations to optimal adaptation by the choice of species and plantation tree density. More thorough testing of the validity

of the maximum NCP principle is therefore needed, especially in natural vegetation, which is in a near-equilibrium, optimal
state with its environment, as can be found in savannas.

Another challenge is found in the definition of the carbon costs, which are key to the calculation of the Net Carbon Profit and, thus, the resulting vegetation properties. Especially the carbon cost parameter for the water transport system remains highly uncertain. This carbon cost parameter could not be estimated based on literature and was instead calibrated by Schymanski et al. (2009b) at a relatively wet savanna site, the Howard Springs eddy covariance site (Fluxnet code AU-How, Beringer
et al., 2016). However, this parameter reflects how optimal vegetation behaviour shapes the plant hydraulic system, and is therefore crucial in order to apply these optimality principles to different sites under different circumstances. Theoretically, this carbon cost factor should be a constant, and transferable to different sites, independent from vegetation species or the climate. Nevertheless, this needs to be tested thoroughly, as several studies suggest that the efficiency of plant hydraulics can depend on environmental conditions such as temperature or water stress (Roderick and Berry, 2001; Mencuccini et al., 2007;
Hacke et al., 2001), indicating that the carbon cost related to the water transport system, which determines the plant hydraulics, may depend on these conditions as well. A correct implementation of the plant hydraulic system is also generally important for terrestrial biosphere modelling, as plant hydraulics are currently being implemented in TBMs as a major limitation for water use during drought (Christoffersen et al., 2016; Sperry et al., 2017; Kennedy et al., 2019). In the VOM, plant hydraulic limitations are not explicitly modelled, but vegetation adjusts the extent of its vascular system based on the water transport
costs as a function of rooting depth and vegetation cover. Therefore, an adequate representation of these carbon costs is highly important.

Through their link with vegetation cover, the water transport costs also influence above-ground phenology, which has a strong influence on the modelled flux exchanges. However, the VOM-application by Schymanski et al. (2009b) predicted a vegetation cover of 100 % at Howard Springs, in contrast with remotely sensed observations. This was explained by a low-bias
in the remotely sensed vegetation due to seasonal lagoons within the grid cell, but applications of the VOM under a wider range of environmental conditions should provide a more thorough evaluation of the correctness of the predicted vegetation cover as a result of the carbon costs and benefits. In general, seasonal dynamics of vegetation cover are hard to capture in terrestrial biosphere modelling (Richardson et al., 2012), but this is highly important regarding the effects of climate change (Piao et al., 2019).
The last category of carbon costs to calculate the NCP relates to the respiration and maintenance of the root system. Thus, maximizing the NCP should lead to a prognostic rooting depth, which reflects the specific long-term situation at a certain site. For example, rooting depths are known to strongly vary with precipitation (Schenk and Jackson, 2002) and are, therefore, likely to change over the precipitation gradient of the NATT (Williams et al., 1996b). In general, a correct representation of the root depths are known to be important for a correct simulation of fluxes (Kleidon and Heimann, 1998; Yang et al., 2016;
Wang-Erlandsson et al., 2016), which is in the VOM linked to a correct representation of the optimality-principles and the accompanying carbon costs and benefits.

Hence, a systematic evaluation of the prognostic, optimality-based vegetation properties under different climatological conditions in comparison with traditional modelling approaches should provide insights about the added value of the optimality

principles implemented in the VOM for simulating dynamic vegetation behaviour. Therefore, we apply the VOM to the same savanna sites that were used in the model intercomparison by Whitley et al. (2016) along the North Australian Tropical Transect (NATT, Hutley et al., 2011). Using this experimental framework, we could test the VOM and the vegetation optimality principles, over a well-defined environmental gradient, and a comparison could be made with several state-of-the-art TBMs that were used by Whitley et al. (2016). We formulated the following hypotheses:

1. The optimality-based model is not substantially worse at capturing the seasonal amplitudes and mean annual values of observed carbon and water fluxes than conventional models analyzed by Whitley et al. (2016) along the NATT.

2. The plant hydraulic system has carbon costs that only depend on the plant size, represented by root depth and vegetation cover, and a constant independent of species and climate (i.e. the water transport cost parameter).

3. The optimality-based dynamic vegetation cover, as a result of maximizing NCP, reproduces the carbon and water fluxes better than a prescribed mean seasonal phenology in the VOM, derived from remote sensing data.

4. The optimality-based constant rooting depth at each site, as a result of maximizing NCP, reproduces the carbon and water fluxes better than a prescribed homogenous rooting depth across all sites in the VOM.

## 2  Methodology

The hypotheses were addressed by setting up the VOM for five study sites along the North Australian Tropical Transect and several numerical experiments were defined, as described respectively in Sect. 2.1 and 2.3. More details about the VOM and the long-term and short-term optimization of the Net Carbon Profit can be found in Sect. 2.2, as well as in the accompanying technical note by Nijzink et al. (2021). All the data pre- and post-processing and model runs were done in an open science approach using the RENKU[1] platform as a tool to make the research reproducible, enable transparent numerical analyses, and track all steps in the scientific process. The entire workflow including code and input data can be found online[2,3], whereas more details of our analyses can be found in Supplements S1-S8.

### 2.1  Study sites

In order to test the VOM across a precipitation gradient and compare to the modelling outputs of Whitley et al. (2016), the same study sites of the North Australian Tropical Transect (NATT, Hutley et al., 2011) were selected. These savanna sites are located between $12.5°S$ and $22.5°S$ and span over a distance of approximately 1000 km (Figure 1 and Table 1) with a mean annual precipitation increasing from the most southern sites towards the north from 500 to 1700 mm/year (Figure 1 and Table 1). The sites are mainly (open-forest) savanna sites with mostly evergreen overstorey (*Eucalypt* species) and understorey grasses (*Sorghum* species). Only Sturt Plains, the most southern and driest site, is not a savanna site, but a natural Mitchell

---

[1]https://renkulab.io/

[2]https://renkulab.io/gitlab/remko.nijzink/vomcases

[3]https://doi.org/10.5281/zenodo.5789101

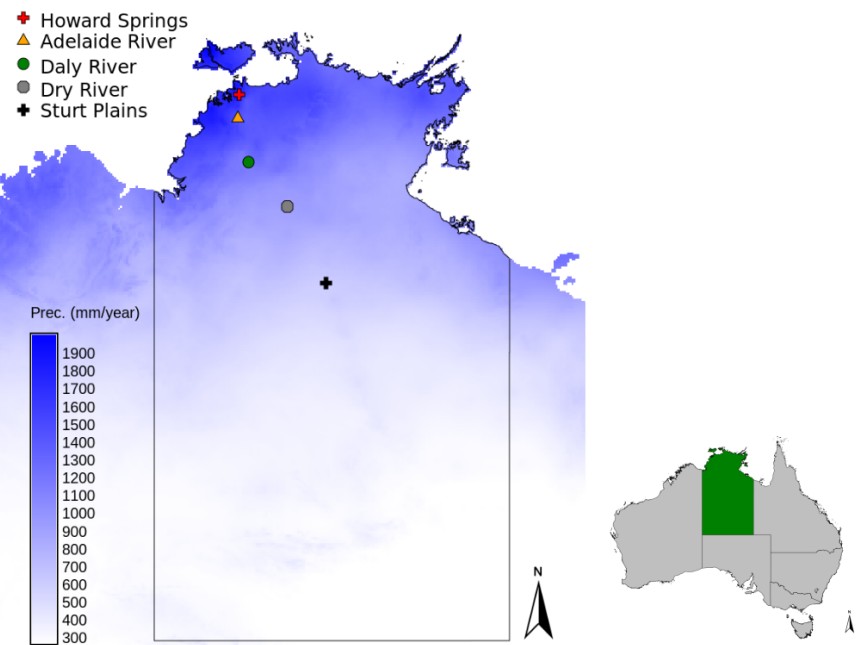

**Figure 1.** Locations of the study sites along the North Australian Tropical Transect in the Nortern Territory of Australia, with the mean annual precipitation shown in the blue colorscale (SILO Data Drill, Jeffrey et al., 2001, calculated for 1980-2017) .

grassland dominated by *Astrebla* grasses (Hutley et al., 2011) and no perennial overstorey species. The contrasting vegetation is due to the soil type that is a defining characteristic of this ecosystem, being a cracking gray vertisol with a silty clay texture. The other sites have soils that mainly consist of sandy loam (Howard Springs, Daly River, Dry River) and silt loam and sandy clay loam (Adelaide River). See also Table 1 for an overview of the different vegetation species and soils at the different sites, more detailed soil profiles are given in Supplement S8, Table S8.3.

## 2.2 Vegetation Optimality Model

The Vegetation Optimality Model (VOM, Schymanski et al., 2009b, 2015) couples a vegetation and water balance model and optimizes vegetation properties in order to maximize the Net Carbon Profit, i.e. the total amount of carbon assimilated from the atmosphere by photosynthesis minus the carbon costs for maintenance and respiration of the plant tissues. The model code and documentation can be found online[4][5] and version v0.5[6] of the model was used in this study. The model is described in more detail in Schymanski et al. (2009b, 2015) and more specifics about the current model set-up can be found in the accompanying technical note of Nijzink et al. (2021). For completeness, a brief description of the model is given below.

---

[4]https://github.com/schymans/VOM

[5]https://vom.readthedocs.io

[6]https://doi.org/10.5281/zenodo.3630081

**Table 1.** Characteristics of the study sites along the North Australian Tropical Transect, vegetation data from Hutley et al. (2011), Hutley (2015) and Whitley et al. (2016), with Eucalyptus (Eu.), Erythrophleum (Er.), Terminalia (Te.), Corymbia (Co.), Planchonia (Pl.), Themeda (Th.), Hetropogan (He.), and Chrysopogon (Ch.). Meteorological data is taken from the SILO Data Drill (Jeffrey et al., 2001) for the model periods of 1-1-1980 until 31-12-2017, with the potential evaporation calculated according to the FAO Penman-Monteith formula (Allen et al., 1998). The ratio of the net radiation $Rn$ with the latent heat of vaporization $\lambda$ mutiplied with the precipitation $P$, is defined here as the aridity $Rn/\lambda P$. Tree cover is determined as the minimum value of the mean monthly projective cover based on fPar-observations (Donohue et al., 2013). The maximum grass cover was found by subtracting the tree cover from the remotely sensed projective cover.

| Study Site | Howard Springs | Adelaide River | Daly River | Dry River | Sturt Plains |
|---|---|---|---|---|---|
| FLUXNET ID | AU-How | AU-Ade | AU-DaS | AU-Dry | AU-Stp |
| Coordinates | 12.49S | 13.08S | 14.16S | 15.26S | 17.15S |
| | 131.35E | 131.12E | 131.39E | 132.37E | 133.35E |
| Prec. (mm year$^{-1}$) | 1747 | 1497 | 1166 | 898 | 616 |
| Pot. evap. (mm year$^{-1}$ ) | 1763 | 1802 | 1896 | 1948 | 2082 |
| Aridity. (-) | 1.03 | 1.18 | 1.48 | 1.87 | 2.70 |
| Net Rad. (MJ m$^{-2}$ year$^{-1-1}$ ) | 4392 | 4313 | 4215 | 4105 | 4079 |
| Mean max. temp. [$^{o}$C] | 37.5 | 38.8 | 40.6 | 41.1 | 43.0 |
| Mean min. temp. [$^{o}$C] | 27.4 | 26.6 | 26.9 | 27.7 | 28.1 |
| Tree cover (%) | 39.8 | 20.8 | 37.5 | 26.6 | 7.4 |
| Max. grass cover (%) | 44.3 | 59.2 | 42.5 | 49.4 | 57.6 |
| Savanna type | Open-forest savanna | Woodland savanna | Woodland savanna | Woodland savanna | Mitchell grassland |
| Soils | Red and gray kandosols | Yellow hydrosols | Red kandosols | Red and gray kandosols | Gray vertisol |
| Flux measurements | from 2001 | 2007-2009 | from 2007 | from 2008 | from 2008 |
| **Species** | | | | | |
| Overstorey | *Eu. miniata* | *Eu. tectifica* | *Te. grandiflora* | *Eu. tetrodonta* | - |
| | *Eu. tetrodonta* | *Co. latifolia* | *Eu. tetrodonta* | *Co. terminalis* | |
| | *Er. chlorostachys* | *Pl. careya* | *Co. latifolia* | *Eu. dichromophloia* | |
| Understorey | *Sorghum spp.* | *Sorghum spp.* | *Sorghum spp.* | *Sorghum intrans* | *Astrebla spp.* |
| | *He. triticeus* | *Ch. fallax* | *He. triticeus* | *Th. Tiandra* | |
| | | | | *Ch. fallax* | |

### 2.2.1 Water balance model

The hydrical schematization consists of a permeable block that contains an unsaturated zone overlying a saturated zone (Schymanski et al., 2015), with an impermeable bed with a prescribed drainage level. Here, the soil is divided into sev-

eral layers, where the matric suction heads and unsaturated conductivities are determined using the water retention model of Van Genuchten (1980). Water can flow between these soil layers based on a discretization of the Buckingham-Darcy equation (Radcliffe and Rasmussen, 2002), that results in the 1-D Richards' equation of steady flow.

The hydrological parameters were set to resemble freely draining conditions, in absence of more detailed information for each of the sites and for consistency with the simulations of Whitley et al. (2016). This was achieved by a total soil thickness of 30 m and a fast drainage parameterization, see for details the accompanying paper of Nijzink et al. (2021). Eventually, precipitation that falls on the soil block can cause immediate surface runoff or infiltration, after which it can be taken up by roots, evaporate from the soils, or drain away at a depth of 30m. The simulations of soil evaporation and vertical fluxes in the

unsaturated zone are described in Schymanski et al. (2008, 2015).

### 2.2.2   Vegetation model

The VOM represents seasonal grass vegetation and perennial trees by two big leaves in the model. The VOM optimizes several vegetation properties dynamically on a short-time scale (daily) and others on a long-term timescale, in order to, eventually, maximize the overall NCP for the entire simulation period (see also Sections 2.2.3, 2.2.4, 2.2.5). Here, photosynthesis was

modelled with a simplified C3 canopy gas exchange model of von Caemmerer (2000), and $CO_2$-uptake is calculated based on irradiance, atmospheric $CO_2$-concentrations, photosynthetic capacity and stomatal conductance. At the same time, root water uptake is calculated based on the differences between root and soil water potentials in each soil layer, using an electrical circuit analogy.

### 2.2.3   Carbon costs and benefits

The maintenance carbon costs for the different plant organs are defined by several cost functions. The carbon costs for the maintenance of the foliage are defined as:

$$R_f = L_{AIc} \cdot c_{tc} \cdot M_{A,p} \tag{1}$$

with $L_{AIc}$ the clumped leaf area (set to 2.5), $c_{tc}$ the leaf turnover cost factor (set to 0.22 $\mu$mol$^{-1}$ s$^{-1}$ m$^{-2}$, see Schymanski et al. (2007)) and $M_{A,p}$ the perennial vegetation cover fraction.

The cost function for the maintenance of the root system is defined as:

$$R_r = c_{Rr} \cdot \left( \frac{r_r}{2} \cdot S_{A,r} \right) \tag{2}$$

with $c_{Rr}$ the respiration rate per fine root volume (0.0017 mol s$^{-1}$ m$^{-3}$), $r_r$ the root radius (set to 0.3*10$^{-3}$ m), see for more details Schymanski et al. (2008) and Nijzink et al. (2021).

The costs for the plant hydraulic system are a linear function of the size of the plant, defined in the VOM by the rooting depth and the vegetation cover:

$$R_v = c_{rv} \cdot M_A \cdot y_r \tag{3}$$

with $c_{rv}$ the cost factor for water transport (mol m$^{-3}$ s$^{-1}$), $M_A$ the fraction of vegetation cover $(-)$, and $y_r$ the rooting depth (m).

Eventually, the Net Carbon Profit is calculated as the total difference in the carbon costs and benefits:

$$NCP = \int \left( A_g(t) - R_f(t) - R_r(t) - R_v(t) \right) dt \tag{4}$$

with $A_g$ the assimilated carbon by photosynthesis, and $t$ representing the time step.

### 2.2.4 Short-term optimization

The seasonal vegetation cover ($M_{A,s}$) and the electron transport capacities at 25 $^o$C for the seasonal and perennial vegetation ($J_{max25,s}$ and $J_{max25,p}$) are allowed to vary on a daily basis in a way to maximize the daily NCP (see also Table S8.2 in Supplement S8). These vegetation properties are optimized on a daily basis by using the actual value and a specific increment above and below this value. At the end of a daily time step, the value that maximized the daily NCP is then kept and used for the next day. This results eventually in a seasonal signal, representing the phenology of the vegetation. The root surface area distributions are adjusted in a way to satisfy the canopy water demand, also on a daily basis.

### 2.2.5 Long-term optimization

A set of vegetation properties is assumed not to vary considerably over the long-term (20-30 years), and optimized for the full simulation period. These vegetation properties involve the rooting depths of the perennial trees and the seasonal grasses, the foliage projected cover of the perennial vegetation, as well as parameters relating to the water use strategies of both the perennial and the seasonal vegetation.

These long-term vegetation properties (see also Table S8.2 in Supplement S8) are optimized for maximum NCP by the Shuffled Complex Evolution algorithm (SCE, Duan et al., 1994) for the entire simulation period of 37 years (1-1-1980 until 31-12-2017). After an initial random seed, the algorithm subdivides the parameter sets into a number complexes (set here to 10). Afterwards, it performs a combination of local optimization within each complex and mixing between complexes to converge to a parameterization that maximizes the NCP.

### 2.2.6 Model input and data

The meteorological data to run the VOM was obtained from the Australian SILO Data Drill (Jeffrey et al., 2001), as it provided long time series (20-30 years) for the optimization of the VOM. The SILO data consists of a gridded dataset with a resolution

of 0.05°, and is based on the interpolation of around 4600 station datasets across Australia. The meteorological data needed for the VOM consists of daily maximum and minimum temperatures, shortwave radiation, precipitation, vapour pressure and atmospheric pressure (see also Supplement S1, Figures S1.1-1.6). Atmospheric $CO_2$-levels were taken from the Mauna Loa
$CO_2$-records (Keeling et al., 2005). This was preferred to using observed values at the flux tower sites due to the required length of the timeseries for the VOM (20-30 years). However, the models run by Whitley et al. (2016) generally used the flux tower meteorological data, and the SILO meteorological data had to be verified with the measured meteorological variables at the flux tower sites. In this analysis, the daily SILO data was replaced for the days that flux tower observations were available, which resulted in an additional uncertainty that remained relatively low (see also Supplement S4 of the accompanying paper
of Nijzink et al. (2021)).

Besides meteorological inputs, the VOM requires soil parameters regarding soil water retentions and hydraulic conductivities for each depth. Field measurements of sand, clay and silt content were used to describe the composition of the soils in the top 10 cm, whereas values for deeper soil layers were taken from the Soil and Landscape Grid of Australia (Viscarra Rossel et al., 2014a,b,c). The resulting fractions of sand, silt and clay allowed to classify the soils into one of the soil textural groups of
Carsel and Parrish (1988), after which the final parameters for the soil water retention model of Van Genuchten (1980) and the hydraulic conductivity could be looked up from the accompanying tables[7] and used in the VOM. The soil compositions in Whitley et al. (2016) would lead to differences in the soil classifications and represented just the upper layers, but these differences remained rather small. Whitley et al. (2016) reported hydraulic conductivities as well, but the models all used their own methods to determine the hydraulic parameters (see Table 2, Whitley et al., 2016). Moreover, the soil hydraulic
conductivities reported by Whitley et al. (2016) were substantially lower than typical values for the reported soil textures (Carsel and Parrish, 1988). At some sites, these low hydraulic conductivities hardly allowed any flow of water and vegetation growth in the VOM. For this reason, for the VOM simulations, soils were solely classified with the field measurements and data of the Soil and Landscape Grid of Australia (Viscarra Rossel et al., 2014a,b,c), whereas the final soil parameters (Table S8.3, Supplement S8) were adopted from Carsel and Parrish (1988).

The flux towers across Australia and New Zealand (OzFlux Beringer et al., 2016) belong to a regional network of the global FLUXNET network and provided time series for the model evaluation. These timeseries consisted of net ecosystem exchange (NEE) of carbon dioxide and latent heat flux (LE), along with incoming and reflected radiation, and meteorological variables usually recorded at OzFlux sites. The flux tower data was processed with the Dingo-algorithm (Beringer et al., 2017), that provided a gap-filled estimation of gross primary productivity (GPP) and LE. LE was converted to evapo-transpiration (ET),
which we define here as the sum of all evaporation and transpiration processes, even though these processes are different in nature (Savenije, 2004). GPP and ET were both used for comparison with the modelled fluxes. The VOM was evaluated for the overlapping time periods of the model time period and the flux tower time period for each site (see Table S8.1 in Supplement S8).

In order to evaluate the foliage cover predicted by the VOM, remotely sensed data of monthly fractions of Photosynthetically
Active Radiation absorbed by vegetation (fPAR) from Donohue et al. (2008, 2013) was used to estimate foliage projected cover

---

[7]see also https://vom.readthedocs.io/en/latest/soildata.html

(FPC) at the different study sites. The maximum possible value of fPAR was defined as 0.95 by Donohue et al. (2008) and relates to maximum projective cover (i.e. FPC = 1.0). At the same time, FPC relates linearly with fPAR-data (Asrar et al., 1984; Lu, 2003), and FPC was thus calculated by dividing the fPAR-values by the maximum value of 0.95.

## 2.3 Modelling experiments and intercomparison

Several sets of model runs were performed in order to address the different hypotheses. The first hypothesis relates to a model intercomparison with Whitley et al. (2016), whereas the second hypothesis was addressed by running a sensitivity analysis for the cost factor of water transport. The third and fourth hypothesis were addressed using simulations where some of the prognostic vegetation properties and behaviour where replaced by prescribed values.

### 2.3.1 Model intercomparison

In a first round of model runs to address the first hypothesis, VOM results were compared to the results of Whitley et al. (2016), who used the terrestrial biosphere models SPA (Williams et al., 1996a), MAESPA (Duursma and Medlyn, 2012), CABLE (Kowalczyk et al., 2006; Wang et al., 2011), BIOS2 (Haverd et al., 2013), BESS (Ryu et al., 2011, 2012) and LPJ-GUESS (Smith et al., 2001) to simulate savannas along the NATT (see also Table 2 Whitley et al., 2016). From these TBMs only LPJ-GUESS uses a carbon allocation scheme to simulate canopy dynamics, whereas the other five models use observed 275 leaf area index values from MODIS to represent vegetation dynamics.

The models of Whitley et al. (2016) use the meteorological observations from the flux towers, except for BIOS2, which uses a gridded input dataset (Bureau of Meteorology's Australian Water Availability Project data set, see also Haverd et al., 2013). Therefore, these models were run for a model period between 2-10 years, depending on the availability of the data at the flux towers. The models generally used different rooting depths, ranging from 2 to 10 meter. Soil sand and clay content were here 280 taken from the Australian Soil Classification (Isbell, 2002), but the models all used their own number of soil layers and soil depths, as well as their own methods to determine the hydraulic soil parameters. Parameters related to leaf biochemistry were based on Cernusak et al. (2011), for the TBMs that needed this data. The model parameters were not optimized, except for BIOS2, that used a model-data fusion process in order to optimize the model parameters (Haverd et al., 2013). See also Table 2 in Whitley et al. (2016) for more details.

Similar to Whitley et al. (2016), the model intercomparison is based on ensemble timeseries of daily GPP and ET obtained from the different models over the entire duration of flux tower observations for each site (see Table S8.1 in Supplement S8 for the overlapping model periods with the flux tower observations). Model performance metrics include relative errors between the mean annual fluxes and the seasonal amplitudes. The study of Whitley et al. (2016) ranked the models against empirical benchmark models, as originally proposed by Abramowitz (2012). This comparison against calibrated empirical 290 models was not performed here in favour of a more detailed analysis of the actual time series to provide more insights into model deficiencies relative to the observed vegetation behaviour, but included in Supplement S7.

### 2.3.2 Sensitivity to the water transport cost factor

The carbon costs for the maintenance of the plant hydraulic system are assumed to be a linear function of the plant size, i.e. root depth and vegetation cover, and a constant, which is defined as the cost factor for water transport ($c_{rv}$, see Equation 3).
Schymanski et al. (2009b) argued that this cost factor is independent of vegetation species or study sites, in order to define the NCP as an optimality principle that is transferable across different ecosystems with different climates and vegetation types.

This cost factor cannot be easily estimated from empirical data and was initially set to 1.2 $\mu$mol m$^{-3}$ s$^{-1}$ by Schymanski et al. (2009b) after a sensitivity analysis for Howard Springs. Later on, after an adjustment of the water balance model and another sensitivity analysis at the same site, it was set to 1.0 $\mu$mol m$^{-3}$ s$^{-1}$ by Schymanski et al. (2015). To assess the effect of
this cost factor on the simulations more rigorously and test the second hypothesis, we ran additional optimizations with values for $c_{rv}$ ranging from 0.2 to 3.0 $\mu$mol m$^{-3}$ s$^{-1}$ for each site along the transect. Here, we analyzed if the value of $c_{rv}$ that best reproduces satellite-derived dry season vegetation cover varies between sites by more than 0.2 $\mu$mol m$^{-3}$ s$^{-1}$. Regardless of the result here, we used a value of 1.0 $\mu$mol m$^{-3}$ s$^{-1}$ for all other simulations in this study, as determined by Schymanski et al. (2015).

### 2.3.3 Predicted and prescribed vegetation parameters

To test hypotheses three and four, regarding the optimality of rooting depths and vegetation cover, we ran model simulations with prescribed rooting depths, vegetation cover or both. The total vegetation cover (perennial and seasonal) was prescribed based on monthly fPAR-data from Donohue et al. (2013), once considering inter-annual variability (i.e. using the actual values of fPAR-based vegetation cover, gap-filled using monthly ensemble means) and once ignoring inter-annual variability (i.e. using
monthly ensemble means of the full model period). This was done in order to assess if inter-annual variability in vegetation cover plays an important role and if that role is captured by optimizing vegetation cover. The perennial tree cover was estimated as the minimum value of the monthly ensemble means, and then subtracted from the remaining timeseries to obtain the dynamic component of the seasonal grass cover. For Sturt Plains, the perennial cover was set to 0, as prescription of perennial cover corresponding to the minimum value of the monthly ensemble means (0.07) did not allow the SCE-optimization to converge.
See Figure S4.1 for the observed and constructed time series of projective cover at each site. Figure S4.1e illustrates that vegetation cover at Sturt Plains reaches almost 0 in many years, but not always in the same month, so that there is no month with an ensemble mean of 0. This shows that the estimation of the evergreen component of cover by taking the minimum monthly ensemble mean may not work well in a climate that lacks a clear seasonality. For the simulation runs with prescribed rooting depths, we used the same value of 2 m for both trees and grasses, similar to the simulations by LPJ-GUESS in Whitley
et al. (2016). For the simulations with both prescribed roots and vegetation cover, we used also the fPAR-based vegetation cover, and again a rooting depth of 2 m for both trees and grasses.

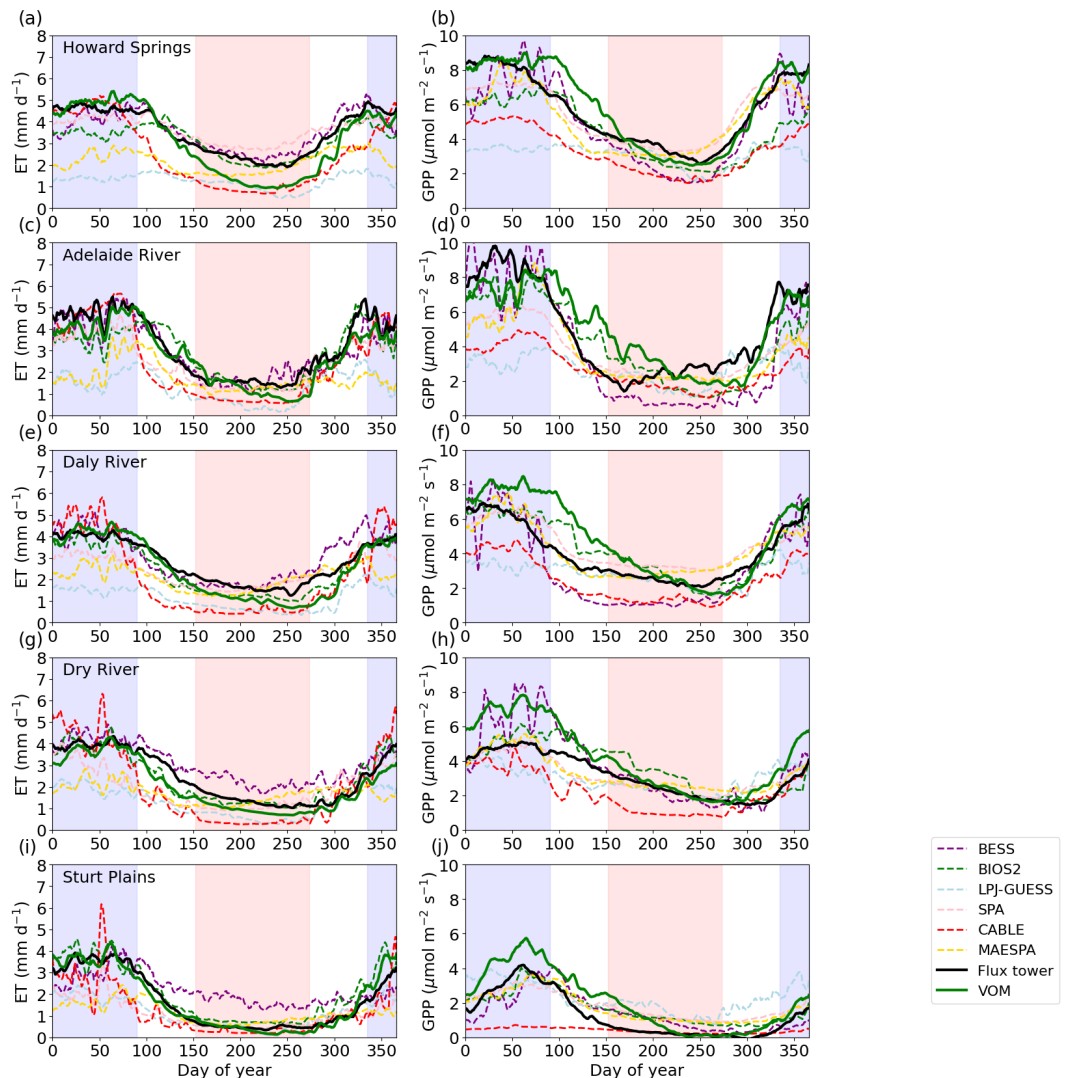

**Figure 2.** Ensemble years of evapo-transpiration (ET) and gross primary productivity (GPP) for the VOM (darkgreen), flux tower observations (black) and the models BESS (dashed purple), BIOS2 (dashed green), LPJ-GUESS (dashed lightblue), SPA (dashed pink), CABLE (dashed red) and MAESPA (dashed yellow) from Whitley et al. (2016), all smoothed by a 7-day moving average. The ensemble years are calculated for the overlapping time periods with the flux tower observations (see Table S8.1 in Supplement S8). The dry season (June-Sept.) is indicated by a red shading, the wet season (Dec.-March) by a blue shading.

# 3 Results

## 3.1 Model intercomparison - Hypothesis 1

The simulations for the model intercomparison (see also Supplement S2) were performed using the same water transport cost
factor for all sites as in Schymanski et al. (2015), i.e. $c_{rv} = 1.0$ $\mu$mol m$^{-3}$ s$^{-1}$. After ranking the models for a set of performance
measures for ET and GPP (correlation coefficient, standard deviation, bias and normalized mean error), the VOM turned out to
have the best average rank, only closely followed by BIOS2 and SPA, see also Supplement S7, Figure S7.3c. More specifically,
these simulations of evapo-transpiration (ET) by the VOM followed the observed seasonality at the different sites relatively
closely, with a tendency to underestimate ET during the dry season (June-September) at some sites, although to a lesser degree
than some of the other models (Fig. 2, left column). Gross primary productivity (GPP), on the other hand, was systematically
overpredicted by the VOM during the wet season (December-March) and/or at the transition between wet and dry seasons,
making it the worst performing model for GPP for limited periods at some sites (Figure 2, right column). However, in terms of
mean annual ET, the VOM belonged to the three best performing models along with BESS and BIOS2 (Figure 3a-b) with on
average an off-set from the observations of 102.8 mm year$^{-1}$ (BESS and BIOS2 with respectively 64.0 and 65.7 mm year$^{-1}$),
whereas the mean annual GPP was better for SPA, MAESPA and BIOS2 with average off-sets of 4.9, 10.9 and 5.0 mol m$^{-2}$
year$^{-1}$, respectively, compared to the VOM with an average off-set of 18.9 mol m$^{-2}$ year$^{-1}$. The remaining models either
strongly underestimated ET or GPP, or both at some sites or throughout the transect. In contrast to the general underestimation
of GPP by the other models, the fully optimized VOM-runs showed a tendency to overestimate GPP along the whole transect
(except for Adelaide River), whereas VOM-runs with prescribed vegetation cover led to correct values of mean annual GPP
at the drier sites, while underestimating GPP at the wetter sites (see Figure 3b). In terms of relative errors in the mean annual
fluxes, the VOM simulations show the most consistently small errors in ET across the transect compared to the other models,
except for BIOS2, with a relative error of less than 0.25 across all sites (Figure 3c, on average -0.1, in comparison with BESS
0.11, BIOS2 -0.03, CABLE -0.27, LPJ-GUESS -0.53, MAESPA -0.34 and SPA -0.17), while the relative errors in GPP increase
with the aridity of the site (Figure 3d) and make the VOM one of the worst performing models in terms of reproducing observed
mean annual GPP a the drier sites, resulting in on average a relative error of 0.25 (BESS -0.05, BIOS2 0.03, CABLE -0.45,
LPJ-GUESS -0.12, MAESPA 0.01 and SPA 0.04).

The VOM is among the models with the smallest relative errors for the seasonal amplitude in ET (Figure 3e, with on average
a relative error of 0.08), except for Howard Springs and Daly River, where the seasonal amplitude is strongly overestimated
mainly due to underestimation of the dry season fluxes. In contrast, most of the other models (CABLE being the exception) tend
to underestimate the seasonal amplitude in ET, which can be noted from the relative errors in the seasonal amplitudes (Fig. 3e,
with on average a relative error for CABLE of 0.57, and BESS, BIOS2 LPJ-GUESS, MAESPA and SPA with an error of -0.06,
-0.06, -0.40, -0.44, -0.20, respectively). In the ensemble time series (Figure 2) it can mainly be noted that the maximum values
in ET of LPJ-GUESS and MAESPA are too low, whereas the minimum values of all the models remain relatively similar. The
seasonal amplitude of GPP was also overestimated by the VOM (except for Adelaide River), but was on average the smallest of
all models (0.09, whereas BESS, BIOS2, CABLE, LPJ-GUESS, MAESPA and SPA all had errors of 0.32, -0.15, -0.37, -0.46,

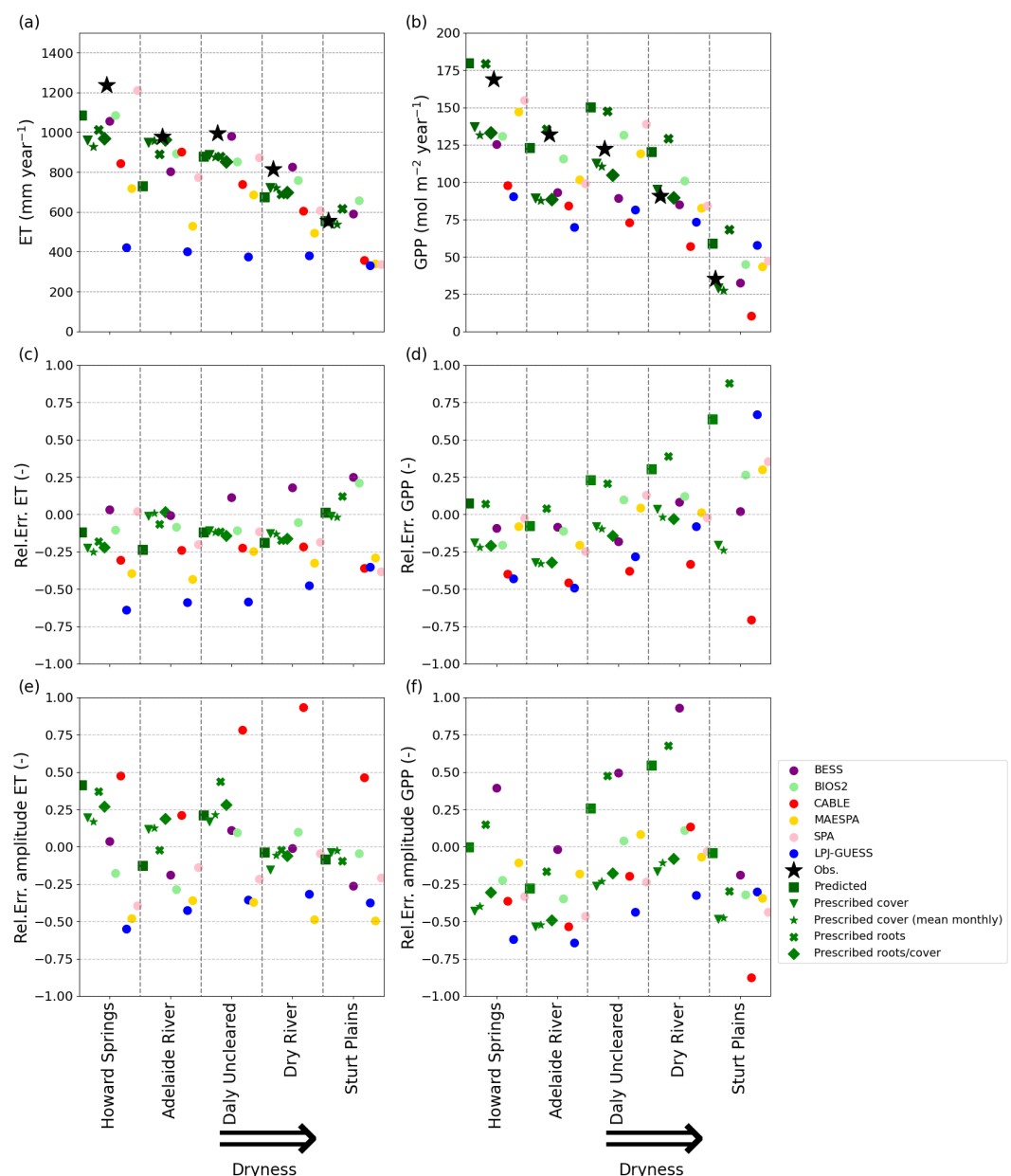

**Figure 3.** Performance of the VOM, the VOM with prescribed cover, prescribed roots, prescribed roots and cover, and the models analyzed by Whitley et al. (2016) in comparison with flux tower observations. a) and b): Mean annual evapo-transpiration (ET) and gross primary productivity (GPP); c) and d): the relative error of the mean annual ET and GPP; e) and f): the relative error for the mean seasonal amplitude of ET and GPP. All metrics for all models are calculated for the overlapping time period of the model time period and the flux tower time period for each site (see Table Table S8.1 in Supplement S8).

-0.12 and -0.30 respectively.) However, the overestimation of the VOM was merely due to overestimated dry season fluxes instead of underestimated dry season fluxes (Figs. 2 and 3f). VOM simulations with prescribed cover largely remove this bias at the drier sites, while leading to opposite errors at the wetter sites. All of the other models underestimate the seasonal signals in GPP, except for BESS, which overestimates the seasonal signal even more than the VOM for most sites (with an average relative error for BESS of 0.32 and for the VOM of 0.09).

## 3.2 Sensitivity to carbon costs of plant water transport - Hypothesis 2

In order to investigate the systematic departures of the VOM simulations from observations, we investigated the sensitivity of the results to the cost factor for water transport $c_{rv}$ (see also Supplement S3). The cost factor has a substantial influence on the fluxes (Figure 4), but especially the vegetation cover during the dry season (i.e. the minimum values in Figure 4c) decrease strongly with an increasing cost factor. At all sites, the vegetation cover is very sensitive to the cost factor for values below 1-1.8 $\mu$mol m$^{-3}$ s$^{-1}$ and becomes much less sensitive at higher $c_{rv}$ values (Figs. 5a-e).

The decrease in persistent vegetation cover with increasing values of $c_{rv}$ was not always smooth in our simulations (Fig. 5a-e). Therefore, when optimizing for the $c_{rv}$ value that most closely reproduces observed dry season FPC (solid lines in Fig. 5a-e) , visual interpolation of the dots would result in a range of candidate values for each site. For example, at Howard Springs, the best value would be somewhere between 0.6 and 1.0 $\mu$mol m$^{-3}$ s$^{-1}$, at Adelaide River anywhere between 1.4 and 2.6, at Daly River between 0.8 and 1.0, at Dry River between 1.0 and 1.2, and at Sturt Plains between 1.4 and 2.0 $\mu$mol m$^{-3}$ s$^{-1}$. Conversely, if we were to choose the same value for all sites along the transect, then a value of 1.4 $\mu$mol m$^{-3}$ s$^{-1}$ would minimize the Euclidian distance based on the error for all five study sites (Figure 5f), closely followed by the value of 1.0 $\mu$mol m$^{-3}$ s$^{-1}$.

## 3.3 Predicted and prescribed foliage cover - Hypothesis 3

The VOM predicted that projective cover would reach 100% during the wet season at all sites, but observed maximum values of cover ranged between 60% and 80% for the different sites (see e.g. Fig. 4c for Howard Springs, see Supplement S2 Figures S2.1, S2.4, S2.7, S2.10, S2.13 for the other sites). At the same time, carbon uptake rates during the wet season were overpredicted at most sites by the VOM (Fig. 2), which prompts the question if the overprediction of projective cover might be linked to the overprediction of carbon uptake. To test this hypothesis, the VOM simulations were repeated with prescribed foliage projective cover based on observed fPAR-data (Donohue et al., 2013), while everything else was optimized as usual (see for the full analysis Supplement S4).

When prescribing fractional tree cover to the value of 0.07 derived from observed dry season cover at Sturt Plains (Supplement S4, Fig. S4.1e), the Shuffled Complex Evolution (SCE) algorithm did not converge, i.e. no parameter sets resulting in positive NCP were found. Only when the tree cover and tree rooting depths were set to zero and all the observed cover was interpreted as grass cover (consistent with the fact that the site is an extensively grazed grassland) the SCE found optimal water use strategy parameters for the grasses, resulting in a positive maximum NCP.

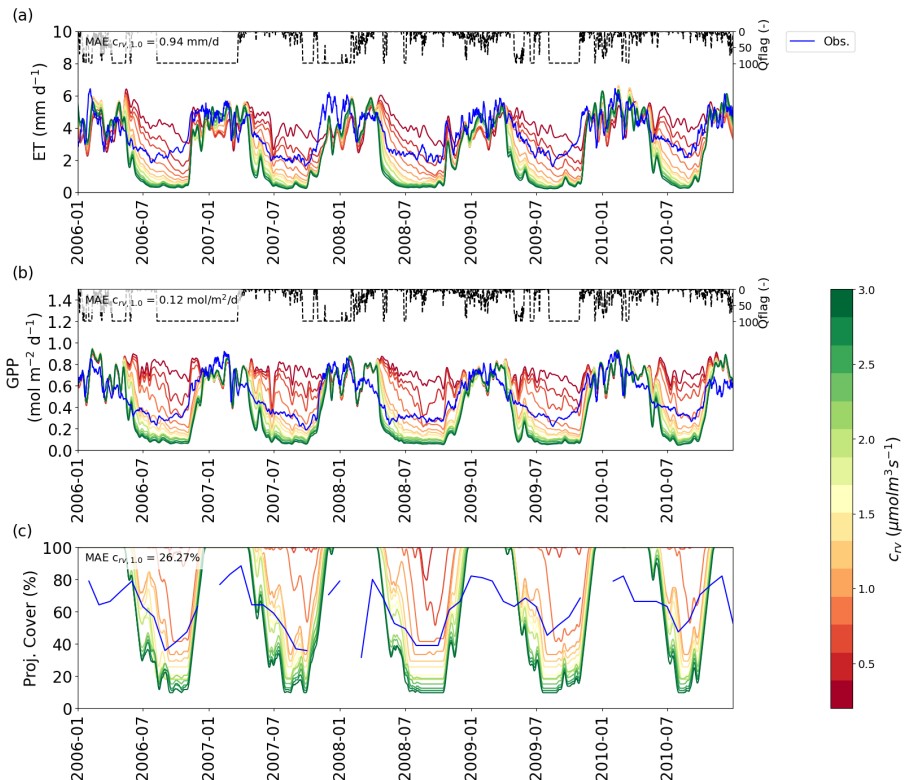

**Figure 4.** VOM-results for different values of the cost factor $c_{rv}$ (color scale) for Howard Springs for 2006-2010 (as a subset from 1980-2017), with a) the evapo-transpiration (ET), with flux tower observations in blue b) gross primary productivity (GPP), with flux tower observations in blue and c) projective cover, with the observed fraction of vegetation cover based on fPAR-data (Donohue et al., 2013) in blue. The time series for ET and GPP are all smoothed with moving average with a window of 7 days. Daily average quality flags for the flux tower observations are added on top and range from 0 (no missing values) to a 100 (completely gap-filled). In each panel, the mean absolute error (MAE) is given for the simulation with $c_{rv} = 1.0$ $\mu$mol m$^{-3}$ s$^{-1}$.

In general, prescribing vegetation cover in the VOM consistently reduced the mean annual GPP at all study sites (Figure 6b), which led to a shift in the average off-set from the observations from +18.9 to -18.7 mol m$^{-3}$ s$^{-1}$. This happens most

strongly during the wet season (Figure 6d), as expected, and during the transition to the dry season (Figure 6h). During this transition from the wet to the dry season (April-May, Figures 6g,h), prescribing cover indeed improved the simulations for all sites, whereas the reductions in the wet season resulted in an underprediction of GPP at all sites (Fig. 6d). Even in the dry season, prescribed cover reduced simulated GPP and improved the match with observations, except for one site (Dry River, Fig. 6f). The effect on evaporation was less pronounced, as reduced cover led to reduced transpiration, but this was partly

compensated for by increased bare soil evaporation (Fig. 6a, c, e, g, i). In fact, simulated soil evaporation was always lowest for optimized rooting depth and vegetation cover, suggesting that optimal vegetation behaviour in the VOM depicts competition

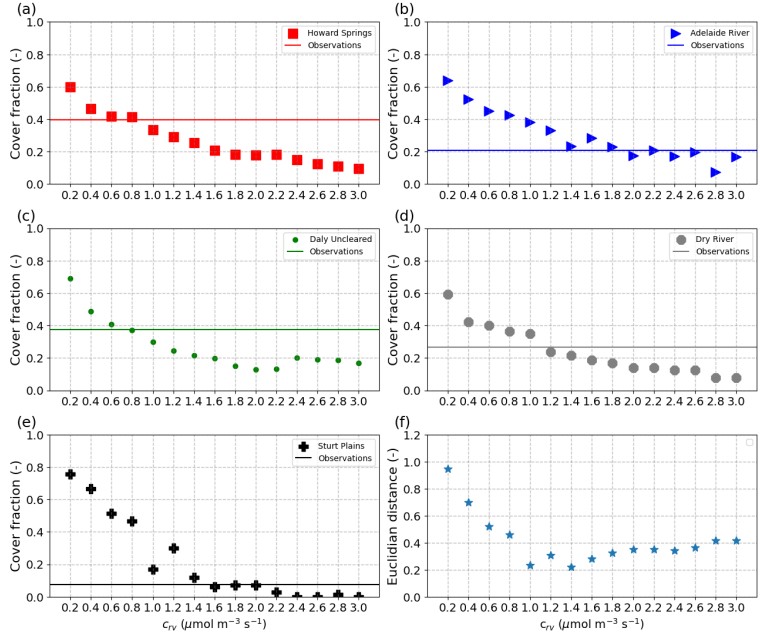

**Figure 5.** Simulated dry season percentage vegetation cover at each site (a-e) and the Euclidian distance of errors between sites (f) as a function of the water transport system cost factor ($c_{rv}$). Symbols illustrate simulated vegetation cover of perennial vegetation ($M_{a,p}$), while solid lines illustrate the perennial tree cover derived from observations, as described in Sect. 2.3.3. The Euclidean distance in panel f) was based on the error between observed and simulated vegetation cover during the dry season for all five study site, i.e. $ED = \sqrt{\sum_{i=1}^{n} E_i^2}$, with $E_i$ the error for each site for a given $c_{rv}$.

for water between vegetation and other processes (Fig. 6a, c, e, g and i). Moreover, the total transpiration of both the perennial and seasonal vegetation was always the highest for the fully optimized VOM.

Looking at the fluxes in more detail and taking Daly River as an example (Figure 7), we see that the initial overestimation of
GPP during the transition from the wet to the dry season is largely eliminated when projected cover dynamics are prescribed based on remote sensing observations, but merely caused by generally lower GPP rates and at the cost of underestimating GPP during the early wet season. Similar patterns can be seen for the other sites in Supplement S4, Fig. S4.2-4.6, except for Dry River and Sturt Plains, where the wet seasons are shifted more towards the middle of the year. At Dry River (Supplement S4, Fig. S4.5), prescribed cover captured wet season GPP much better than predicted cover, while it overestimated dry season
fluxes compared to observations and simulations based on predicted cover. The plots in Supplement S4 also illustrate that the two different ways of prescribing vegetation cover, with and without inter-annual variability, only led to marginal differences in the simulated fluxes.

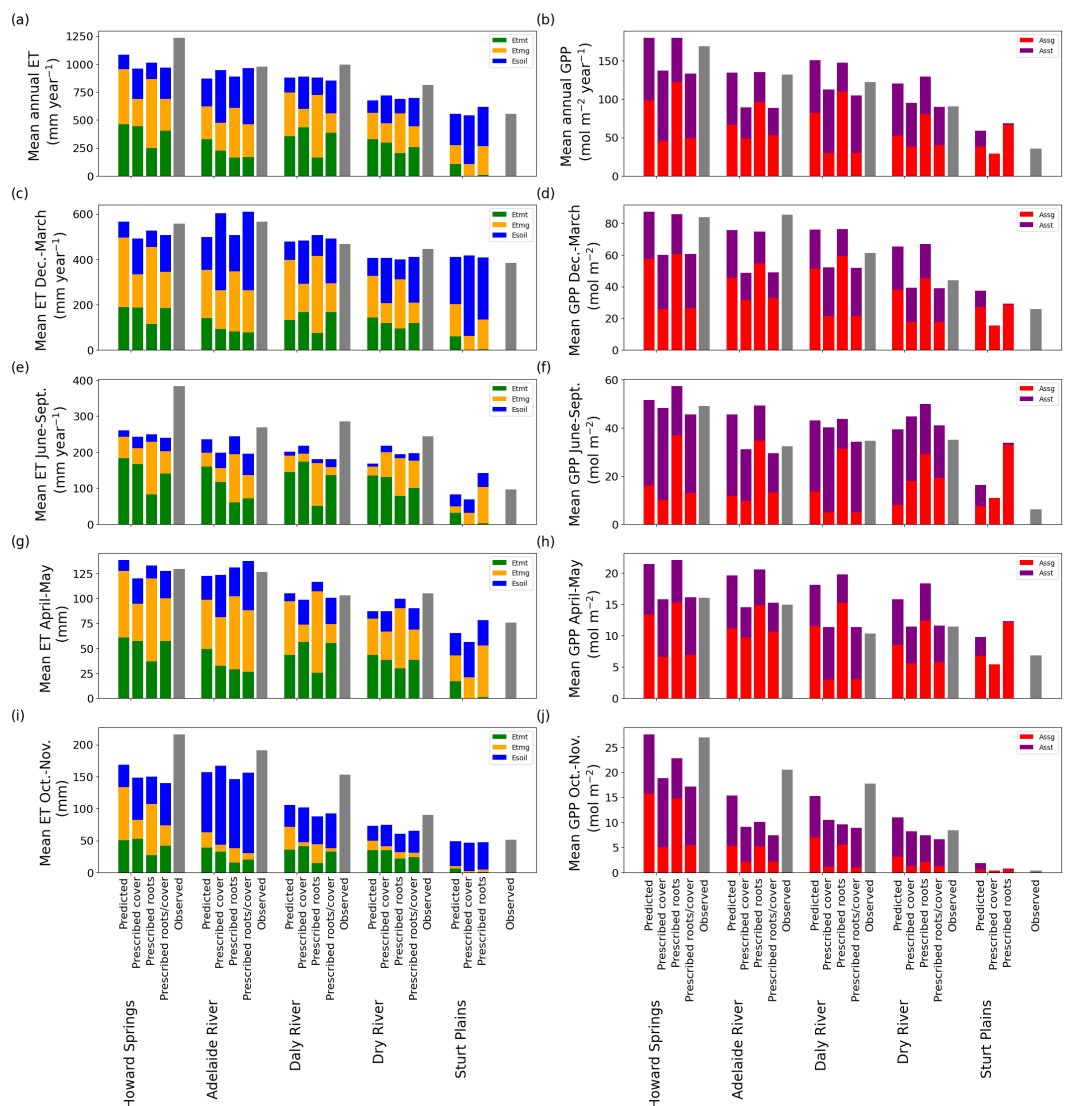

**Figure 6.** Observed and predicted fluxes of water and $CO_2$ using prognostic or prescribed cover, prescribed roots, or prescribed roots and cover, at the different sites. Simulated evapo-transpiration (ET) was sub-divided into soil evaporation (green), tree transpiration (orange) and grass transpiration (blue), while simulated gross primary productvity (GPP) was subdivided into that of trees (red) and grasses (purple). Observations are shown as gray bars. Panels c) and d) represent the fluxes for December-March (generally the wet season), e) and f) the flux partition for June - September (i.e. generally the dry season), and g) and h) the fluxes for April-May (i.e. generally the transition from the wet to the dry season), and i) and j) the fluxes for October-November (i.e. generally the transition from the dry to the wet season). All metrics for all models are calculated for the overlapping time period of the model time period and the flux tower time period for each site (see Table S8.1 in Supplement S8).

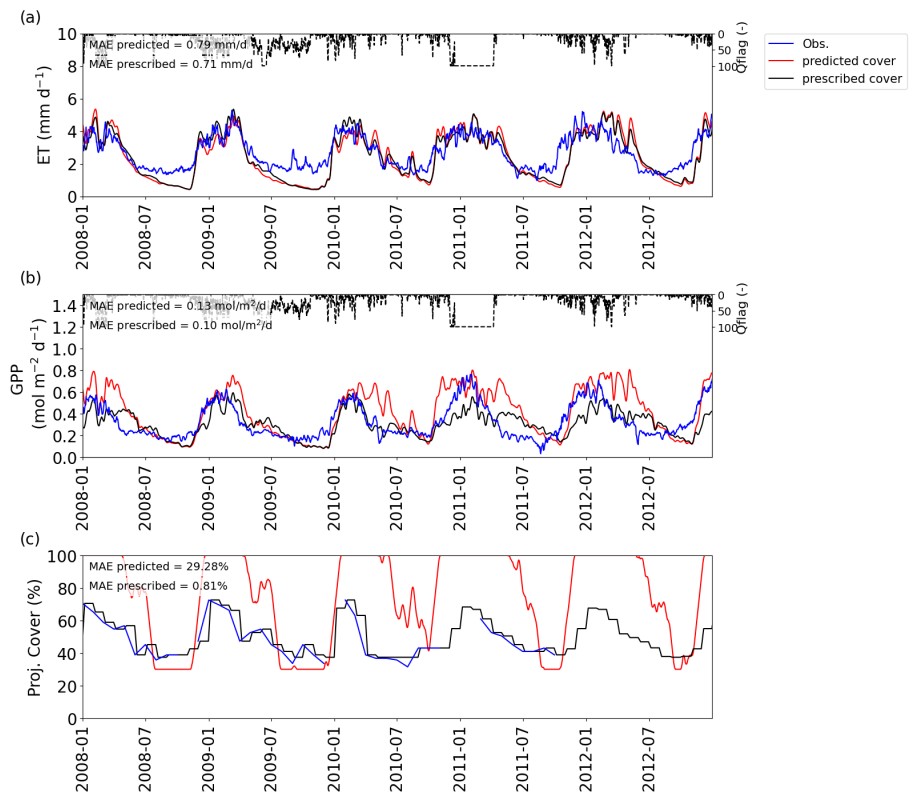

**Figure 7.** Comparison between the VOM with prescribed and predicted vegetation cover for Daly River for 2008-2012 (as a subset from 1980-2017), for a) evapo-transpiration (ET) and b) gross primary productivity (GPP) and c) projective cover, with model results obtained by the predicted cover in red and the prescribed cover in black. The flux tower observations and fPAR-derived projective cover are shown in blue. Daily average quality flags for the flux tower observations are added on top and range from 0 (no missing values) to a 100 (completely gap-filled). The time series for ET and GPP are all smoothed with a moving average with a window of 7 days.

### 3.4 Predicted and prescribed rooting depths - Hypothesis 4

To assess if optimization of rooting depths in the VOM has any effect on predicting fluxes, we compared the original simula-
tions, that optimized rooting depths, with simulations based on prescribed rooting depths. The optimized grass rooting depths were all similar at around 0.5 m over the entire transect, and tree rooting depths varied between 0.6 and 2.2 m (Figure 8a and b), whereas rooting depths of 2 m for trees and grasses alike were prescribed in the new simulations. Even though there was a decreasing pattern of the optimized tree rooting depths over the transect (shallower roots towards drier sites), prescribing rooting depths did not lead to substantially different mean annual evapo-transpiration and gross primary productivity in com-
parison with the optimized VOM (Figure 3a and b, Figure 6a and b) with deviations from the observed mean annual values of respectively -102.8 mm year$^{-1}$ and 97.8 mm year$^{-1}$. Only mean annual GPP slightly increased for the drier sites if rooting depths of 2 m were prescribed. However, there are clear effects of prescribing 2 m rooting depth at the seasonal scale. The

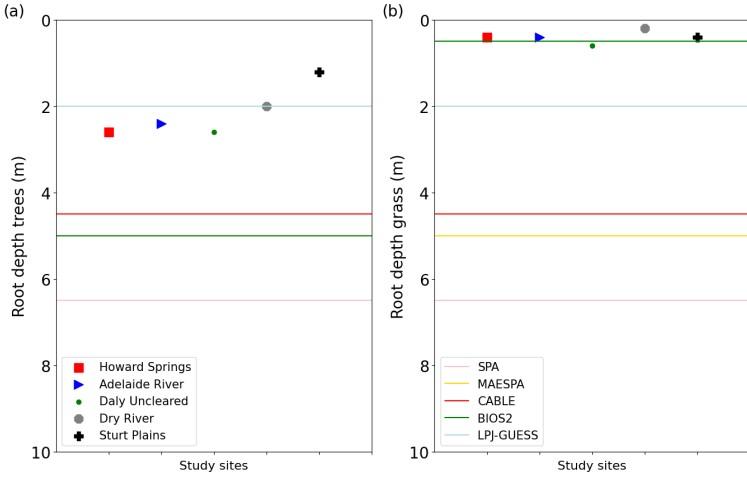

**Figure 8.** Modelled rooting depths for a) trees and b) grasses, for Howard Springs (red squares), Adelaide River (blue triangles), Daly River (green circles), Dry River (gray octagon), and Sturt Plains (black cross). The colored lines represent the rooting depths of the models used in Whitley et al. (2016) with SPA (blue), MAESPA (orange), CABLE (green), BIOS2 (red) and LPJ-GUESS (purple).

prescribed rooting depths result in increased transpiration and GPP rates in the early dry season, followed by strongly reduced fluxes in the late dry season, compared with simulations based on optimized root depths, as can be seen in Figure 9 for Dry

River, as an example, but as also illustrated for the other sites in Supplement S5, Figures S5.1-5.5. Figure S3.8 in Supplement S3 indicates a decreasing sensitivity of the predicted rooting depths to the cost factor for water transport $c_{rv}$ with decreasing mean annual rainfall along the transect. For example, predicted tree rooting depths at Howard Springs ranged between 1 and 8 m when $c_{rv}$ was varied by an order of magnitude, whereas Sturt Plains only had a variability between 3 m and 1 m for the same range of $c_{rv}$. Similar to fractional cover (Fig. 5), the predicted rooting depths became less sensitive to $c_{rv}$ above a value

of around 1-1.5 $\mu$mol m$^{-3}$ s$^{-1}$.

### 3.5   Predicted and prescribed rooting depths and vegetation cover - Hypotheses 3 and 4

Prescribing both rooting depths and vegetation cover in the VOM (see also Supplement S6) resulted in relatively similar results to those obtained by prescribing vegetation cover only (Figure 6). The differences stay relatively small for ET, in comparison with the fully prognostic VOM (Figure 6a), but there are larger differences for GPP (Figure 6b).

The reductions in GPP by prescribing roots and vegetation cover are mainly caused by differences in the dry season, and in the transition from the dry to the wet season (Figure 6d, h). As a result, the relative errors in GPP and the relative error of the seasonal amplitude in GPP are reduced (Figure 3d, f), showing again that the prescribed vegetation cover corrects the overestimation in GPP by the VOM. At Sturt Plains, the driest site, the VOM did not produce any solutions with positive NCP, indicating that the combination of observed vegetation cover and prescribed rooting depth employed here is not possible if the

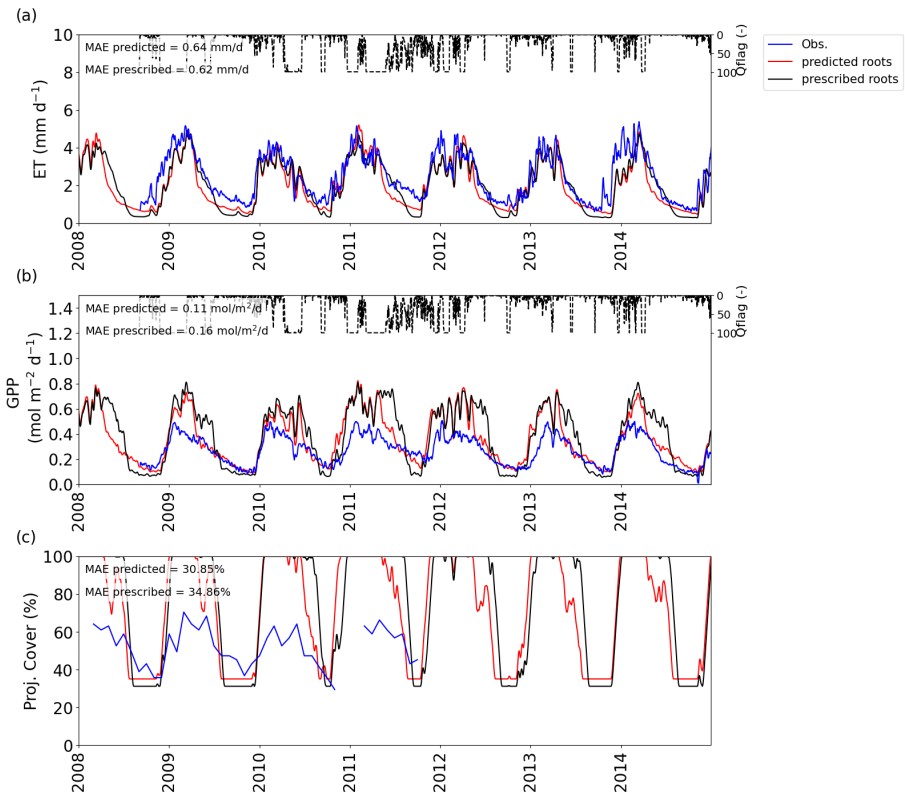

**Figure 9.** VOM-results for prescribed (black) and predicted (red) rooting depths for Dry River for 2008-2013 (as a subset from 1980-2017), with a) the evapo-transpiration (ET), with flux tower observations in blue b) gross primary productivity (GPP), with flux tower observations in blue and c) projective cover, with the observed fraction of vegetation cover based on fPAR-data (Donohue et al., 2013). Daily average quality flags for the flux tower observations are added on top of a) and b) and range from 0 (no missing values) to a 100 (completely gap-filled). The time series for ET and GPP are all smoothed with moving average with a window of 7 days.

carbon costs and benefits prescribed in the VOM are realistic (missing bars for "Prescribed roots/cover" at Sturt Plains in Fig. 6). If only one was prescribed, the VOM was able to find a solution by adequately adjusting the other to achieve positive NCP.

## 4  Discussion

The results illustrate that the VOM reproduced observed water vapour and $CO_2$ fluxes much better than the only other model with prognostic phenology (LPJ-GUESS), and not significantly worse than any of the other models. However, our study also
revealed some persistent shortcomings of the VOM and/or the underlying optimality principles. Possible reasons for these shortcomings and potential ways forward will be discussed below, along with the hypotheses formulated in the introduction.

## 4.1 Model comparison

In comparison with the models tested at the same sites by Whitley et al. (2016), the VOM reproduced observed water vapour and $CO_2$ fluxes clearly better than the only other model with prognostic phenology (LPJ-GUESS), and although it was out-performed in certain aspects by one or another model, its combined performance in simulating ET and GPP ranked on top in the model inter-comparison (Fig. 7.3c in Supplement S7). This is surprising, considering that most of the other models used observed seasonal variations in vegetation cover and other site-specific vegetation input, while LPJ-GUESS was the only other model that predicted vegetation dynamics. LPJ-GUESS, however, was not able to reproduce the seasonal amplitude of the fluxes at all and severely underestimated mean annual fluxes throughout the transect, except for an overestimation of GPP at the driest site. According to Whitley et al. (2016), the failure of LPJ-GUESS to capture observed seasonality can be explained by how carbon is allocated in the model on an annual basis, in addition to the empirical prescription of phenology. In contrast, the VOM predicted phenology dynamically from day to day, and had resulting fluxes that showed a much more pronounced seasonal signal, which also corresponded more to the observed flux tower observations.

## 4.2 Effect of hydrological setting

Our results suggest that the hydrological setting at the different sites, especially the position of the groundwater table, influence the performance of the VOM. The model underestimated evapo-transpiration (ET) during the dry season at all sites, especially in the late dry season. At Howard Springs, this was linked to the lack of a groundwater table in the present simulations, which were based on the assumption of freely draining conditions, for compatibility with the previous model intercomparison by Whitley et al. (2016). Previous model applications of the VOM (Schymanski et al., 2009b, 2015), which simulated a variable groundwater table based on local topography at Howard Springs, did not suffer from such a strong underestimation of dry season fluxes (see the accompanying technical note of Nijzink et al. (2021)).

This emphasizes the importance of a correct hydrological parameterization and confirms previous findings that groundwater can have a strong influence on the land surface fluxes (e.g. York et al., 2002; Bierkens and van den Hurk, 2007; Maxwell et al., 2007). It is likely that the role of the hydrological parameterization is more important for the wetter sites along the transect, as greater input of precipitation during the wet season implies a greater potential for soil moisture carry-over into the dry season. Unfortunately, groundwater data was not available in the close vicinity of the other sites of the transect, even though boreholes near the drier sites of Daly River (approx. 2 km distance), Dry River (approx. 13 km) and Sturt Plains (approx. 10 km) suggested deeper groundwater tables for these sites (Supplement S2, Figures S2.8, S2.11, S2.14).

## 4.3 Carbon costs of the plant hydraulic system

We found that different values of the cost factor for water transport ($c_{rv}$) could be chosen for each site to best reproduce observed dry season FPC, but these values would not at the same time improve the match with observed mean annual fluxes, and in fact, for most sites, different $c_{rv}$ values would need to be chosen to reproduce either mean annual ET or GPP (Supplement S3, Fig. S3.1). However, the value of $c_{rv}$ affects dry season fluxes much more than wet season fluxes (Figure 4), so adjusting

$c_{rv}$ to match observed mean annual fluxes would mean that an overestimation of fluxes during the wet season would be compensated by an underestimation during the dry season. Furthermore, comparison between the optimum $c_{rv}$ for Howard Springs in this study (0.6-0.8 $\mu$mol m$^{-3}$ s$^{-1}$) with the value of 1.0 $\mu$mol m$^{-3}$ s$^{-1}$ obtained by Schymanski et al. (2015), who considered groundwater influence, suggests that the variation in the best $c_{rv}$ between sites may also be due to different levels of misrepresentation of the hydrological settings. For example, the neglect of a potentially existing groundwater influence in this study would result in under-estimated vegetation cover in the dry season, which could be compensated for by lowering $c_{rv}$. Similarly, an overestimated soil water holding capacity in the model would result in overestimated dry season vegetation cover (points shifted upwards in Figure 5), which would lead to the choice of a higher value of $c_{rv}$ to compensate.

More generally, the sensitivity of the simulations to the value of $c_{rv}$ emphasizes the need for a more thorough understanding of the carbon costs related to the water transport infrastructure and their dependence on environmental conditions, including effects of temperature or water stress on the efficiency of plant hydraulics (Roderick and Berry, 2001; Mencuccini et al., 2007; Hacke et al., 2001). At the same time, it could be argued that this cost factor may be different for seasonal grasses and perennial trees, akin to differences in carbon costs for short-lived and long-lived leaves (Eamus and Prichard, 1998).

Note that the value of 1.0 $\mu$mol m$^{-3}$ s$^{-1}$ chosen by Schymanski et al. (2015), and also in the present study, turned out to lead to the best combined reproduction of dry season vegetation cover along the transect, similar to the value of 1.4 $\mu$mol m$^{-3}$ s$^{-1}$, as indicated in Fig. 5f. Essentially, we found that the sites with the highest observed dry season cover, Howard Springs and Daly River, would be best represented with the same low value of 0.8 $\mu$mol m$^{-3}$ s$^{-1}$, whereas the sites with lower dry season cover would be best represented by higher values between 1.2 and 1.8 $\mu$mol m$^{-3}$ s$^{-1}$. Neither the simulated nor the observed dry season vegetation cover values are clearly related to mean annual rainfall (Table 1 and Fig. 5), suggesting that the length of the dry periods and the storage capacity for plant-available water originating from the wet periods likely play an important role in determining dry season vegetation cover. Hence, it is very important not to hide a wrong representation of this storage capacity in the model by a site-specific, tuned value of $c_{rv}$.

## 4.4 Predicted foliage cover

Using the same cost factor for the water transport system ($c_{rv}$) across all sites, the predicted perennial vegetation cover fraction was between 0.3 and 0.4 at all sites except for the driest, grassland site, where it was still predicted at almost 0.2. This is in contrast to the observed dry season foliage projected cover (FPC), which ranged between 0.07 at the driest and roughly 0.4 at the wettest site (Figure 5), where Adelaide River was an outlier with only 0.2 FPC in the dry season. The strong over-prediction of perennial vegetation cover at the driest site, Sturt Plains, might relate to a low sensitivity of the NCP to perennial cover at this site and the fact that it is a lightly grazed grassland site (Hutley et al., 2011). The NCP-values are generally low here (see also Supplement S4, Figure S4.11), and there is only a small difference in NCP between the predicted optimal tree cover of 16.9% and a forced perennial cover of 0.0 (921.56 and 726.9 mol m$^{-2}$ respectively, Supplement 4, Fig. S4.11). VOM-simulations with prescribed seasonal cover and a forced tree cover of 0.0 resulted in an even smaller NCP of 397.7 mol m$^{-2}$, while prescribing perennial cover with 7% and seasonal vegetation cover with the remaining observed cover yielded only negative NCP-values. Due to the large inter-annual variability at this site and relatively short period of time with observed cover (see Supplement

S4, Fig. S4.1e), for the largest part of the modelling period the prescribed cover just represented the long-term mean monthly values of fPAR-based observations, which is likely far off the actual values. Note also that a nearby ungrazed woodland site had a dry season leaf area index (LAI) of 0.4 (Hutley et al., 2011) which is roughly consistent with the predicted fractional cover of 0.2 and assumed clumped LAI within vegetated patches of 2.5 (Schymanski et al., 2009b).

Another issue is that the VOM consistently overestimates vegetation cover in the wet season at all sites, always reaching 100%, which is not consistent with observations. Schymanski et al. (2009b) already pointed out that observed vegetation cover at Howard Springs did never reach 100% and explained this by seasonal lagoons within the remote sensing grid cell, implicating a low-bias in the remotely sensed vegetation cover. However, in the present study we found that observed vegetation cover never reaches 100% at any of the five sites, while the simulations always do. At the same time, an overprediction of GPP at the drier sites was removed when prescribing vegetation cover based on observations, rather than predicting it based on optimality principles (Figure 6). This suggests that there might be a fundamental bias in the VOM's representation of the costs and benefits of foliage, and thus the resulting NCP-values.

In the VOM, the relation between leaf area and vegetation cover is assumed to be linear, but this relationship is highly non-linear in reality (Choudhury, 1987). Due to this linear representation of a non-linear function, the carbon costs for high values of vegetation cover are likely underestimated in the VOM, which could be the reason for overestimated wet season vegetation cover. However, note that prescribed vegetation cover led to an underestimation of GPP at the wetter sites, so there is likely still an issue with the translation between the modelled vegetation cover and that derived from remotely sensed fPAR.

Nevertheless, prognostic vegetation dynamics, i.e. phenologies, are often incorporated in terrestrial biosphere models based on other principles than the optimality-based method here. For example, a large number of vegetation models still use empirical, temperature-based estimates of leaf dynamics (Basler, 2016; Piao et al., 2019), or empirical estimates based on a selection of multiple biophysical variables, such as daylength, evaporative demand or temperatures (Jolly et al., 2005). Interestingly, our results seem to be in line with the ecological optimality hypotheses of Eagleson (1978, 1982) that minimize water stress and maximize transpiration. Nevertheless, only a few models use carbon cost and benefits (Kikuzawa, 1991) in order to derive leaf dynamics. As mentioned above, refinements are likely needed in the representation of carbon costs and benefits of water transport and light attenuation in the VOM, but the results presented here are extremely encouraging for the inclusion of optimality in vegetation models, as it does enable the reproduction of the observed seasonal signal in phenology and fluxes equally well to models that rely on phenological data as input.

## 4.5 Predicted rooting depths

Rooting depths predicted by the VOM decreased with decreasing mean annual precipitation along the transect. This is consistent with the findings of Schenk and Jackson (2002), who, based on a large global dataset, also suggested that rooting depths decrease with decreasing mean annual precipitation. In fact, the empirical relation between rooting depths and mean annual precipitation proposed by Schenk and Jackson (2002, Fig. 7), when extrapolated to the mean annual precipitation range of 600-1700 mm/year for the NATT sites, would predict rooting depths between 1.5 and 3.4 m, which is relatively close to the range predicted by the VOM (1.2-2.6 m Figure 8a, or up to 4.2 m when considering groundwater influence in Schymanski

et al. (2015)). This suggests that optimizing vegetation parameters for NCP enables predicting a realistic spatial variation in rooting depths. Note, however, that rooting depths are species-dependent and should be expected to also depend on local hydrological conditions, most notably the distance to the water table (Kollet and Maxwell, 2008). Hence, any misrepresentation of the hydrological settings in the VOM would result in a biased prediction of rooting depths. Whitley et al. (2016) argued that transpiration rates were maintained relatively high during the dry season due to the presence of deep roots, which is in line with studies suggesting that Eucalypt trees in Northern Australia are deep rooted and, therefore, maintain high transpiration rates during the dry season (O'Grady et al., 1999; Eamus et al., 2000). As discussed above, the position of the water table was not captured in this set of VOM simulations, so we should not expect that the model would accurately predict rooting depths at the different sites.

However, the predicted rooting depths of grasses were more or less constant along the transect, which contrasts with the changing understorey species composition along the transect. At Howards Springs, the understorey is dominated by annual *Sorghum* grasses, whereas a reduction of *Sorghum* grasses and a shift towards more perennial grasses (e.g. *Astrebla spp.*) exists with declining precipitation amounts over the transect, see also Table 1 in Ma et al. (2013) and Hutley et al. (2011). In addition, the predicted grass rooting depths at most sites were sensitive to the cost factor for water transport ($c_{rv}$), similarly to tree rooting depths (Fig. S3.7f and h). This confirms the importance of a better understanding of these costs for our ability to correctly predict rooting depths and vegetation cover.

In line with other studies that showed that rooting depths need to be accurately represented (Kleidon and Heimann, 1998; Yang et al., 2016; Wang-Erlandsson et al., 2016) in order to improve modelled fluxes, we hypothesised that optimized rooting depths will result in a better reproduction of fluxes than prescribed, homogeneous rooting depths along the transect. To test this hypothesis, the VOM simulations were run with 2 m rooting depth at each site, which is the same rooting depth as assumed for the LPJ-GUESS runs in Whitley et al. (2016). Note that the optimized tree rooting depths (Figure 8a) are in most cases larger than the prescribed values of 2 m, which helps the plants in the VOM to maintain transpiration and GPP during the dry season. At Howard Springs, for example, values drop much lower in the dry season for both evaporation and gross primary productivity when roots are prescribed (Supplement S5, Figure S5.1). Optimized grass rooting depths, in contrast, are much shallower (ca. 0.5 m) than the prescribed 2 m. This results in reduced overall transpiration and gross primary productivity during the early dry season when grasses are still active, compared to simulations with prescribed rooting depths at most sites. At Dry River (Figure 9), where predicted and prescribed tree rooting depths were similar, it becomes obvious that the prescribed 2 m rooting depths for grasses enhance total transpiration during the early dry season at the cost of severely reduced transpiration in the late dry season, when the trees seem to be running out of water. Interestingly, at this site, both simulations underestimate dry season evapo-transpiration, one in the early dry season, the other in the late dry season. In general, however, predicted optimality-based rooting depths improve simulations of dry season GPP over the transect (see Supplement S5, Figure S5.6f).

The results also reveal that predicted rooting depths result in less biased simulations of optimal foliage projected cover (see e.g. Figure 9c). If rooting depth was prescribed to 2 m, full cover was maintained longer into the dry periods, which led to a strong overestimation of GPP in Apr-May (Fig. 6h, Fig. 9, Figs. S5.1-S5.5). However, if both rooting depths and foliage cover were prescribed, GPP resembled closely those of prescribed foliage cover (Fig. 6h), indicating that the rooting depths mainly

affect foliage cover and to a lesser extent the fluxes directly. Therefore, when introducing optimality principles in vegetation modelling, it is important to capture optimization of both rooting depth and foliage cover simultaneously. This is also confirmed by the different rooting depths that are found after prescribing just the vegetation cover (see also Figure S4.9 in Supplement S4).

The optimized rooting depths are mostly smaller than those prescribed in the different models studied by Whitley et al. (2016) (Fig. 8a-b), but the rooting depths varied also substantially between the different models. This variability is striking, especially as new methods have led to improved estimates of rooting depths or root zone storage capacities, for example by water balance based estimations (Gao et al., 2014; Wang-Erlandsson et al., 2016; Nijzink et al., 2016) or extended databases (Schenk et al., 2009). Some authors also used optimality approaches for simulating rooting depths, e.g. by maximizing net primary productivity in a simple biosphere/soil hydrology model (Kleidon and Heimann, 1998; Hwang et al., 2009), or transpiration (Collins and Bras, 2007), or maximum long-term root water uptake (van Wijk and Bouten, 2001) in a hydrological model. Guswa (2008, 2010) considered a prescribed root length density and determines the optimal rooting depth as the one where a further increase in rooting depth incurs more carbon costs than the additional carbon gain related to greater water storage, calculated using a bucket model and prescribed water use efficiency. Speich et al. (2018) found that this method allowed reproduction of rooting depths deduced from flux data at a range of temperate and cold eddy covariance sites in Europe, but they were not able to evaluate it for Mediterranean sites, as their data-driven estimation of rooting depths failed. From this perspective, our finding that optimal rooting depths resulted in improved simulations of water vapour and $CO_2$ fluxes in savanna ecosystems along a precipitation gradient is a valuable addition to optimality research and can serve as further motivation to implement optimality in global scale TBMs, such as the implementation of optimality-based vertical root distributions in the LSM Noah-MP (Wang et al., 2018).

## 4.6 Data quality constraints

Eddy covariance systems cannot generate reliable flux data during rainfall events, which results in large proportions of gap-filling, predominantly during the wet season at the sites investigated here (see data quality flags in Figures 4, 7 and 9, and in Supplement S2). However, if we only consider days with less than 50% gap filled data, our main findings remain valid. For illustration purposes, we summarized some of the results in seasonal sums of fluxes, which did include gap-filled data (Fig. 6), but the general conclusions can be confirmed by considering the original time series data with less than 50% gap filling proportion. The length of the flux datasets varies strongly between sites, with the shortest being only 2 years at Adelaide River and the longest at Howard Springs, where we analyzed 16 years of flux data (Fig. S5.1). Another constraint is the quality of meteorological forcing at the sites. In order to obtain 20 years of meteorological forcing for each site, we used gridded, interpolated data of the Australian SILO Data Drill (Jeffrey et al., 2001). In the accompanying paper by Nijzink et al. (2021), we did not find significant differences in the simulated fluxes at Howard Springs if the meteorological data from the flux tower was used to run the VOM instead. However, this does not exclude that there are potential biases in the gridded data extracted for the other sites.

 **5  Conclusions**

Below, we address the original hypotheses of our study, followed by general conclusions.

1. The optimality-based model is not substantially worse at capturing the seasonal amplitudes and mean annual values of observed carbon and water fluxes than conventional models analyzed by Whitley et al. (2016) along the NATT.

   To our surprise, the maximum Net Carbon Profit principle enabled the Vegetation Optimality Model (VOM) to capture the seasonal amplitudes and mean annual values of carbon and water fluxes along the North Australian Tropical Transect (NATT) to a similar or better degree than conventional models. The VOM showed a strong seasonal signal, in contrast to several other models that generally underestimated the seasonal amplitude of the fluxes. This is remarkable and promising, considering that many vegetation properties are predicted by the VOM, such as tree cover, rooting depth and grass phenology, which have to be prescribed in most of the other models. Therefore, we can conclude that optimality-driven models provide a promising way to model savanna ecosystems.

2. The plant hydraulic system has carbon costs that only depend on the plant size, represented by root depth and vegetation cover, and a constant independent of species and climate (i.e. the water transport cost parameter).

   Observed variations in dry season foliage projected cover could be reproduced better if the water transport cost parameter was varied between 0.8 and 2.0 $\mu$mol m$^{-3}$ s$^{-1}$, but this would not necessarily lead to improved reproduction of observed fluxes. However, the variation in this parameter indicates that more factors play a role in defining these carbon costs. For example, we found that the representation of hydrological conditions plays an important role here (see below), but it can be argued that temperature and water stress have an influence on these carbon costs and the resulting plant hydraulic system as well. Therefore, we can neither confirm nor reject this hypothesis at present.

3. The optimality-based dynamic vegetation cover, as a result of maximizing NCP, reproduces the carbon and water fluxes better than a prescribed mean seasonal phenology in the VOM, derived from remote sensing data.

   In the fully prognostic mode, the VOM overestimated GPP during the transition from wet to dry seasons at all sites, which was corrected if vegetation cover was prescribed based on observations. This indicates a rejection of the hypothesis, but, on the other hand, observation-based cover underestimated GPP at the wetter sites in all seasons, leading to a worse bias at these sites than in the fully prognostic model runs. The predicted vegetation cover showed similar dynamics to the observed, but was also systematically overestimated during the wet season (reaching 100% at all sites). At least, this points at a necessary improvement in the carbon costs and benefits of the foliage cover, but the promising seasonal signal indicates that these merely involve refinements. Eventually, this hypothesis can not be fully accepted nor rejected, but we also gained new insights by this (see below).

4. The optimality-based constant rooting depth at each site, as a result of maximizing NCP, reproduces the carbon and water fluxes better than a prescribed homogenous rooting depth across all sites in the VOM.

The optimality-based prognostic rooting depths reproduced carbon and water fluxes better than a prescribed, homogeneous rooting depth of 2 m for all sites. Predicted tree rooting depths were generally deeper than the prescribed, leading to less decrease in the fluxes during the dry season in comparison with prescribed rooting depths. At the same time, predicted grass rooting depths (<0.5 m) were much shallower than the prescribed and reproduced the observed decay in grass fluxes better. The predicted values of tree rooting depths also decreased with mean annual precipitation as observed elsewhere in the literature, indicating that the optimality principles were able to provide realistic predictions of rooting depths. Hence, this hypothesis was accepted.

Based on these hypotheses, we were able to pinpoint systematic shortcomings in the applied optimality theory that indicate a path to additional research promising to substantially improve our capability to predict responses of savanna systems to environmental change:

- One of these shortcomings relates to the representation of the site-specific hydrological conditions. Due to lack of information about local drainage conditions, the model was parameterized as free draining, resulting in underestimated dry season water use. Further research into a better representation of groundwater dynamics at such sites would likely improve our ability to simulate dry season fluxes. Naturally, this argument has been made by others in the literature before, but the application of optimality theory reveals the sensitivity of vegetation behaviour to the hydrological setting even more strongly, as in this model, vegetation adapts its rooting depth, root distribution and foliage cover to the interplay between the local climate and hydrology.

- Another shortcoming relates to the understanding of trade-offs related to the plant water transport infrastructure. The transport of water from deep soil layers upwards and its distribution over the foliage requires a sophisticated water transport infrastructure, which is likely linked to substantial carbon costs for its construction and maintenance. We found that the simulations of fluxes especially in the dry season are very sensitive to the parameterization of these costs and more research into their quantification and relation to environmental conditions will likely further improve our modelling capabilities of savanna systems based on maximizing the NCP.

- A third shortcoming relates to the representation of carbon costs with respect to foliage. The linear relationship between leaf area and absorbed radiation in the current model, in combination with the neglect of leaf reflectivity, results in underestimated carbon costs of maintaining a canopy that can absorb all the light and hence overestimated the fraction of absorbed radiation and GPP in the wet season. The results of the VOM with both prescribed vegetation cover and rooting depths remained close the results of the VOM with just prescribed vegetation cover. In other words, the influence of a prescribed vegetation cover was much stronger compared to the influence of prescribed rooting depths. This shows that especially improvements in the costs and benefits of the vegetation cover may be needed in order to improve the optimality theory applied here.

We conclude that the optimality-based model performed surprisingly well in comparison with conventional models, which also use more empirically-based vegetation properties. The optimality-based model has the promise of performing equally

well when predicting vegetation responses to yet unseen conditions, as its underlying principles are unlikely to change as the environment changes. Therefore, this optimality-theory provides an alternative for parameterizing vegetation properties in TBMs, which are often still prescribed by observations, derived by carbon allocation schemes, or taken from plant functional types. Even though the theory is incorporated in the VOM, the Net Carbon Profit is a straightforward objective, defined here by the total assimilation of $CO_2$ by photosynthesis, and the carbon costs for the root system, the plant hydraulic system, and the foliage. Hence, future directions could also include the application of the theory to other TBMs, with slightly different definitions of the internal processes, in order to verify also the generality of the theory.

In addition, the independence of the model from calibration and local observations as input, by using the optimality theory, enabled us to discover systematic biases and room for improvement that would otherwise be obscured by adjusting parameters to local conditions. Hence, the use of an objective, independent measure for the parameterization of TBMs shows already advantages, even though we identified several possible refinements of this measure here.

We conclude that maximization of the Net Carbon Profit shows great promise for optimizing vegetation parameters in TBMs. Optimality theory can enable a more systematic evaluation of TBM performance, clearer identification of misrepresented processes, and lead to an improved understanding of vegetation behaviour and its prediction for future climate scenarios.

*Code and data availability.* Model code is available on github (https://github.com/schymans/VOM) and the full analysis including all scripts and data are available on renku (https://renkulab.io/gitlab/remko.nijzink/vomcases). Static versions of these repositories can be found on zenodo.org, for the VOM-v0.5 (https://doi.org/10.5281/zenodo.3630081) and the renku-repository (https://doi.org/10.5281/zenodo.5789101).

*Author contributions.* SJS and RN designed the set-up of the study. Model code was originally developed by SJS, but updated and modified by RN. RN did the pre-processing, modelling and post-processing. LH and JB provided site-specific knowledge and data. The main manuscript was prepared by RN, together with input from SJS. LH and JB provided corrections, suggestions and textual inputs for the main manuscript.

*Competing interests.* The authors declare no conflict of interest

*Acknowledgements.* This study is part of the WAVE-project funded by the Luxembourg National Research Fund (FNR) ATTRACT programme (A16/SR/11254288).

We would like to acknowledge Rhys Whitley, Vanessa Haver, Martin de Kauwe and Longhui Li for providing the data from Whitley et al. (2016).

This work used eddy covariance data collected by the TERN-OzFlux facility (http://data.ozflux.org.au/portal/home). OzFlux would like to acknowledge the financial support of the Australian Federal Government via the National Collaborative Research Infrastructure Scheme and the Education Investment Fund.

We acknowledge the SILO Data Drill hosted by the Queensland Department of Environment and Science for providing the meteorological data (https://www.longpaddock.qld.gov.au/silo/).

We acknowledge the Scripps CO2 program (https://scrippsco2.ucsd.edu/data/atmospheric_co2/primary_mlo_co2_record.html) for the Mauna Loa Observatory Records.

We also acknowledge CSIRO for the Soil and Landscape Grid of Australia (https://aclep.csiro.au/aclep/soilandlandscapegrid/index.html) and the Australian monthly fPAR derived from Advanced Very High Resolution Radiometer reflectances - version 5 (https://data.csiro.au/dap/landingpage?

We acknowledge the Northern Territory Water Data WebPortal for the groundwater data (https://water.nt.gov.au/).

We would like to thank the two anonymous reviewers, Yuting Yang and the editor Anke Hildebrandt for their constructive feedback and comments, which were very helpful to improve the manuscript.

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
