# Peer review of "Does maximization of net carbon profit enable the prediction of vegetation behaviour in savanna sites along a precipitation gradient?"

_Hydrology and Earth System Sciences, 2021_

## Author Comment (AC1)

**Response to Referee #1**

We would like to thank Referee #1 for the review, which we see as very helpful. The Referee brings forward several valid points that we will improve on in a revised version of the manuscript. Below, we address the comments of Referee #1, with the referee comments written in italics.

*However, into details, I am not satisfied with the manuscript organization and the writing itself. Overall, I found there are many information currently included in the manuscript is unnecessary. The introduction is too long and contains many individual studies, which should be largely shorten with more highly-summarized findings/conclusion from existing individual studies. The detailed site description is also not needed, simply summarize the five sites with their specific properties listed in Table1. Table 2 is suggested to move to supplementary.*

We will shorten the introduction, and condense it, with less individual studies, and more general findings. Similarly, we will also shorten the site descriptions. We will also move Table 2 to the supplementary information.

*On to content, I think the part of dealing with water transport cost parameter is more or less deviates from the main line. I would suggest remove the second hypothesis but describe how this parameter was chosen (either prescribed following previous studies or locally parameterized) in the manuscript. Then, the overall structure of the manuscript become: 1) test the VOM using site observations and compare it with TBMs; 2) what happened if remotely sensed vegetation cover was used? 3) what happened if prescribed rooting depth was used. Followed by discussion. I understand that the water transport cost parameter is also related to the overall performance of the model, but if that is included, why not other model parameters? And also you will need to describe you model in detail to allow readers who do not familiar with the model understand the role of this parameter in the model.*

The analysis of the water transport cost parameter was included, as this is the only parameter in the model, which could not be based on literature values so far. It was originally tuned to achieve reasonable results at only one of the sites, Howard Springs, and hence we find it important to investigate in how far the same value of this parameter can be used at the other sites along the transect. This is crucial for assessing the optimality theory, which is the main goal of this study. We will however explain the importance of this analysis more in the revised manuscript, and the necessity of this analysis for the interpretation of the other analyses.

We also believe that our current structure is not much different as the Referee suggests: 1) test the VOM using site observations and compare it with TBMs 2) find the underlying reasons for deviations with a) what happens if we vary the unknown parameter for the water transport cost? b) What happens if remote sensed vegetation cover was used? c) What happens if prescribed rooting depth was used? As the water transport cost parameter is the most uncertain in the entire model, it only makes sense for us to assess this parameter first, and then expand the analysis in order to find more underlying reasons for deviations from the observations.

Regarding the model details, we would like to refer to the accompanying technical paper in GMD (https://doi.org/10.5194/gmd-2021-151), as well as the previous papers with more details about the VOM.

*Another question is why not include a scenario that consider both prescribed vegetation cover and rooting depth, in comparison to the scenario with both vegetation properties optimized.*

This is a good idea, we will conduct the suggested simulation and present it either in the main paper or in the SI, depending on the importance of the insights gained by it.

*Other comments:*

*Line 215ï1⁄4infiltrate -> infiltration*

Will be changed accordingly.

*Line 216: Why 30m? Is this the defined soil depth in the model? Not sure if the choice of this depth impacts the modelling results.*

The 30m was chosen in order to represent freely draining conditions, with deep groundwater tables. See also the accompanying technical paper in GMD, where we found there is a strong influence. We will also clarify this here.

*Line 225-230. This may present a source of uncertainty, as the observed fluxes are directly linked to the observed meteorological forcing at the sites, whereas the SILO data was used here to inform the model. Suggest to at least evaluate the used SILO data at each site against site-observed meteorological variables during their overlapping periods.*

Please see Supplement S4 of our accompanying technical paper in GMD (https://doi.org/10.5194/gmd-2021-151). We found that replacing the daily meteorological data from SILO, with aggregated daily data from the flux towers, did not lead to strong differences in the results. We will clarify this here in the main manuscript.

*Line 223 Is there any published paper supporting this? Otherwise, simply states this information is measured at each site.*

We believe the referee means Line 232, where we will clarify the source of the soil data.

*Line 260-263 and the following sections. If I understood correctly, the last two hypotheses are related to replacing vegetation properties with prescribed values and the second hypothesis is about water transport cost parameter. Please check.*

We will correct this.

*Line 246 and throughout: evapotranspiration is often written without a hyphen.*

We are aware of this, but this is done on purpose. We try to emphasize here, that it actually involves two different processes: evaporation and transpiration. Hence, this is merely an abbreviation. See also our statements in lines 249-251.

*Line 316-317: Where is the evidence for this? Figure 3. Please indicate where needed. In addition, in figure 2 at Howard Springs (but not for all other sites), there is a light green curve indicating the results of Schymanski 2015. What is the difference is model configuration between Schymanski 2015 and this study? And what is this for? It is not introduced.*

This can be seen in Figure 3b, we will add a reference in the text.

We will also remove the lines related to Schymanski et al. (2015), as this analysis was moved to the accompanying technical paper in GMD.

*Line 400. This is an overstatement. Looking at Figure 2, the VOM considerably overestimates GPP from observation and even compared with other TBMs.*

We will rephrase this, we tried to explain in this paragraph that the VOM did not substantially worse or better than any of the other models.

*Line 500. I do not agree with this hypothesis/statement. This may simply caused by the fact that the adopted VOM was not able to reproduce the actual rooting depth using the embedded optimality principles. Many previous studies have already demonstrated the importance of accurately representing rooting depth in the hydrological model to improve the modelled fluxes, for example, those by Kleidon and Heimann (1998) and more recently by Wang et al. (2016) and Yang et al. (2016).*

*Kleidon, A., and M. Heimann (1998), A method of determining rooting depth from a terrestrial biosphere model and its impacts on the global water and carbon cycle, Global Change Biol., 4(3), 275–286.*
*Wang-Erlandsson, L., W. G. M. Bastiaanssen, H. Gao, J. J€agermeyr, G. B. Senay, A. I. J. M. van Dijk, J. P. Guerschman, P. W. Keys, L. J. Gordon, and H. H. G. Savenije (2016), Global root zone storage capacity from satellite-based evaporation, Hydrol. Earth Syst. Sci., 20(4), 1459–1481.*
*Yang, Y., R. J. Donohue, T. R McVicar (2016), Global estimation of effective plant rooting depth: Implications for hydrological modeling. Water Resources Research, 52, 8260-8276.*

We fully agree that it is important to accurately represent rooting depths, but we formulated the alternative hypothesis in our paper, which we intended to reject. We will formulate this part more clearly and also discuss the suggested references.

---

## Author Comment (AC2)

**Response to Referee #2**

We would like to thank Referee #2 for the review, and will improve on the points brought forward by the referee. Below, we address the comments of Referee #2, with the referee comments written in italics.

1. Hypothesis 1: One of my major concern concern it that the TBMs (or LSMs) in Whitley et al. (2016), which are designed for global-scale, long-term climate projectoin models, are optimized to make a fair comparison to VOM, though they have more specific vegetatioin parameters.

We did not re-run any of the TBM simulations of Whitley et al. (2006) in order to compare to the VOM, we re-used the model results of Whitley et al. (2016), where the TBMs were applied independently of the VOM. Of course, the modellers involved in the study of Whitley et al. (2016) applied different levels of site-specific parametrisation, and we do not know in how far any parameters were optimized in the original study. With this hypothesis, we mainly try to assess where our optimality-based model stands, and if it still gives satisfactory results in comparison with models that were applied in what the specific modellers considered the "best" way possible.

2. Hypothesis 2 is too model specific. Does any other TEMs use this paramter? or what would other models learn from this? or does this hypothesis have any implication for plant adaptation or optimality?

Plant hydraulics is currently being implemented in TBMs as a major limitation for water use during drought, whereas in the VOM, it is only represented in the form of the water transport cost factor. The intention of this hypothesis is to test whether a general water cost factor across sites leads to reasonable results or if site-specific cost factors would yield significantly improved results. Therefore, assessing this cost factors is also highly necessary in order to assess the general concept of the optimality theory, as applied here. We will clarify this in the revised version.

3. Hypothises 3 is also model specific. It depends on model representations of the links between LAI, vegetation cover, light transfer and absorption, assimilated carbon allocation. This mansusript does not clear describe how these processes are represented in VOM.

We will clarify in the revised manuscript that the prognostic simulation of phenology (LAI and vegetation cover dynamics) is a central concern of vegetation models and optimality theory. We also assess here how the optimality theory, as applied in the VOM, can be improved, and therefore we should also assess the prognostic simulation of the projective cover. As requested, we will explain in more detail how phenology is simulated in the VOM and hence provide a better context for this hypothesis. Thank you for pointing out this gap.

4. Hypothesis 4 shoul be the focus of this manuscript. If this would have been well tested, it is enough to be a good paper. However, this would be greatly affected by the rerepsentation of subsurface soil moisture profile and further likined to capillary fringe. I am wondering why the authors did not use the results of Schymanski et al. (2015). So I strongly suggest to include groundwater effects.

We agree with the referee that groundwater effects are important, but none of the TBMs treated in Whitley et al. (2016), considered them, so we did not in this paper, either. However, we would like to refer to the accompanying technical note in GMD (https://doi.org/10.5194/gmd-2021-151), where we

more thoroughly discuss the effects of the groundwater tables and present a systematic comparison with the results of Schymanski et al. (2015).

5. It is not clear to me (I have to read also Schymanski et al., (2015)) that how VOM is optimized. "Maximizing the NCP"? what is the maximum NCP? how do we know the maximum NCP? Please expand Section 2.2.4 a bit to describe this process in model detail.

Thank you for pointing out this lack of clarity. We will add more details in this section. More specifically, the SCE-algorithm samples the long-term vegetation properties. With this, the VOM is run, and over the full period, the total CO2-assimilation minus the total carbon costs represents the Net Carbon Profit. The vegetation properties that lead to the highest NCP, after sampling and going through the full parameter space with the search algorithm, are kept and considered as the optimal vegetation properties.

6. Section 2.3.3, the cost factor for water transport (crv) should be described here. If not reading Schymanski et al., (2008), I did not understand its meaning.

We will add a specific paragraph describing the cost factor for water transport. In lines 122-129, we introduced the challenges related to this cost factor but we will also explain more in 2.3.3.

7. Conclusions: please generalize these conclussions through discussions. I do not care much about how VOM is better or not or how to improve it but more about how to implement improved understandings through VOM studies into the current TBMS that arewidely used in IPCC climate projections.

Point taken. We will re-write the conclusions from the perspective of general benefits of optimality modelling, and add more general implications of our findings. We will emphasize the identified deficiencies and possible improvements of the VOM, and relate them to the general understanding of the applied optimality theory.

---

## Author Comment (AC3)

**Response to Referee #3**

We would like to thank Referee #3 for the review, and will improve on the points raised by the referee. Below, we address the comments of Referee #3, with the referee comments written in italics.

*In general, I felt the authors could have been a bit more careful with their writing. For example, they confuse their hypotheses throughout the manuscript (see below specific comments) which sometimes makes it hard to follow their argumentation. The introduction is a bit too long and, in parts, not very well linked. I liked how the authors explicitly point out the four hypotheses they aim to explore, however, it is not quite clear to me how hypotheses 2 and 3 emerge from the introduction. It further would be nice to have more details on the model description – you could move table 2 to the supplement for example to make more room. Your discussion is detailed but I'd like to see more about possible future directions.*

We are sorry that the numbering of hypothesis was confused a few times and thank the reviewer for picking this up. We will carefully check this in the revised manuscript. We will also improve the introduction by shortening it and improving the flow, as requested.
We tried to introduce the motivation for Hypotheses 2 and 3 in the introduction (Lines 122-129), but will be more explicit in the revised manuscript. Briefly, plant hydraulics are currently being implemented in TBMs as a major limitation for water use during drought, whereas in the VOM, it is only represented in the form of the water transport cost factor. For this reason, we test with this hypothesis whether a general water cost factor across sites leads to reasonable results or if site-specific cost factors would yield significantly improved results.
Regarding Hypothesis 3, we will emphasize more in the introduction that the prognostic simulation of phenology (LAI and vegetation cover dynamics) is a central concern of vegetation models and optimality theory. For this reason, a systematic evaluation of the prognostic vegetation cover in comparison with more conventional approaches (i.e. prescribing values), should show how the optimality theory, as applied in the VOM, extends the capability of conventional models and how it can be further improved. We will more explicitly discuss future directions in this context as well, as requested.
We included a detailed model description in the accompanying manuscript in GMD (https://doi.org/10.5194/gmd-2021-151), which is why there are not too many details in this manuscript. However, also based on the comments of the other referees, we will add more information here and refer more explicitly to this technical note for details.

*Specific comments*

*Line 41-44 The sentence is very long and hard to follow*

We will rephrase and shorten this sentence.

*Line 61 Doesn't the default version of LPJ-GUESS have more than five plant functional types?*

Thank you for pointing this out, we will correct this.

*Line 91-92 The contents of the sentence are not linked very well*

We will rephrase this sentence.

*Line 98 Increase or decrease in annual rainfall?*

We will add here that it is a decrease towards the south.

*Line 118 'optimizing vegetation properties to maximize the NCP'?*

Will be changed accordingly.

*Line 136 Do timescales of precipitation matter? I.e. is annual PPT driving the rooting depth or are seasonal timescales more important?*

We will add more discussion about this topic. In the VOM, rooting depths are a result of the long-term optimization of the roots, and do not only depend on climate but also on hydrology, i.e. the water storage capacity of the soil and the distance to groundwater.

*Line 136 Therefore instead of but? '[...] therefore is likely to change over […]'*

We will change this to "and is therefore likely to change over…".

*Line 162 In table 1 it's AU-How*

We will correct this.

*Line 173 In table 1 it's AU-DaS*

We will correct this.

*Line 199 '[...] is defined by maintenance respiration, projected cover to the turnover and maintenance of leaf area' – I find this sentence a bit unclear*

We will split this sentence into two sentences: " *is defined by maintenance respiration. At the same time, the cover is linked to the turnover and maintenance of leaf area, while...*"

*Line 208 How can seasonal vegetation cover vary on a daily basis? Maybe rephrase*

We will clarify that in the VOM, seasonal vegetation cover is allowed to vary slightly from day to day, resulting in a seasonality with a maximum during the wet season, and a minimum during the dry season.

*Line 225-230 Does SILO provide point data or are the site met data derived from a spatial dataset (if yes which resolution?) I understand the argument that a longer timeseries helps to run the model, but it would be nice to see any sort of comparison between the observed met data at the site and the SILO dataset. I guess in general I would just like to have more information about the input forcing to get an idea about the uncertainty. Do the models from the Whitley paper run with the same meteorological forcing or do they use the data collected at the site?*

We will add more details about the SILO-data. We will also refer to our accompanying technical paper in GMD (https://doi.org/10.5194/gmd-2021-151, Supplement S4), where we replaced the daily meteorological data from SILO with aggregated daily data from the flux towers. Eventually, this did not lead to strong differences in the results. The models in Whitley et al. (2016) were generally run with the flux tower data, except for BIOS2 that also used a gridded product. We will explain this in the text.

*Line 235-236 Can you provide a bit more detail about the water retention model? It was never mentioned before*

We will add more details about the water retention model. There are also more model details in the accompanying technical note in GMD (https://doi.org/10.5194/gmd-2021-151).

*Line 247 Can you specify what the 'usual energy fluxes' are*

We referred here to observations of incoming and reflected solar radiation. This will be clarified.

*Line 249 LE has already been introduced two sentences earlier*

We will correct this.

*Line 264 Isn't the last hypothesis about rooting depth?*

We are sorry for the confusion, we changed the order in a previous version of the manuscript. We will carefully check this and correct it. Thank you for pointing this out.

*Line 265-276 I might have just overlooked in your submission – but can you describe in more detail what the model set up is for the model intercomparison you use from the Whitley et al paper? Surely there will be more detail in the Whitley paper to help understand but while reading your submission I was for example wondering whether there are changes to some of the parameters in the models to capture the site specifics better or whether they 'just' ran in their original configuration with the meteorological forcing from the sites […]*

We will add more specifics about the models here.

*Line 277 Third and fourth hypotheses?*

Thank you, we will correct this.

*Line 294 In the introduction it says -0.1-0.1 for the cost factor for water transport (but I might have understood?)*

In the introduction, we discussed the variation, instead of the absolute value. We will clarify this.

*Line 297 second hypothesis (also Line 298)*

Will be changed accordingly.

*Line 300 'Regardless of the result here' can you explain why you make this decision?*

We used this value, at it was the outcome of the sensitivity analysis by Schymanski et al. (2015). The assumption was that this cost parameter is valid with this value for all sites, while here we assess the original assumption using a sensitivity analysis. We will clarify this a bit more in the revised.

*Figure 2: Maybe include shaded areas for dry and wet season, but also include dry and wet season months in caption. Can you include what the ensemble years are too? Panel e says Daly Uncleared but it was referred to as Daly River before*

Thank you for these suggestions, we will make corrections accordingly. Daly River and Daly Uncleared refer to the same site, we will make this more consistent throughout the manuscript.

*Line 306 Can you define dry/ wet season (which months)?*

The wet season is from December-March, and the dry season from June-September. We will add this.

*Line 327 Not sure, it looks like the minimum is quite similar for the models but for the maximum values, LPJ-GUESS and MAESPA seem to be too low*

Thank you for pointing this out, we will more carefully formulate this sentence.

*Line 400 I'm not sure I agree with this. It may be true when you look at the annual values in figure 3 but based on figure 2 you can't really reach this conclusion.*

We will rephrase this, our main point is that the VOM does not perform substantially worse than the other models.

*Figure 3: Why are the data points connected in panel a and b? It already is hard to distinguish the data points, the lines make it even harder. You also do not connect them in the other panels – it might be nicer to be consistent. Further, it might be helpful to offset the points in a and b (like in c-f). You could also increase the size of the observation marker, it gets lost in all the other points. Lastly, it could be helpful to include an arrow below the lower x-axes indicating whether the sites go from dry to wet or the other way around (but a lot of this is personal reference of course)*

We tried to assess the patterns over the transect, which is why we added the line. But we agree that it may be clearer without the lines, so we will re-plot the figures without lines and then decide which is clearer. We also like the other suggestions, and will add an arrow indicating the dryness and increase the size of the observation markers. Thank you for the suggestions.

*Figure 4 and 7: Maybe use a global legend and remove obs legend from panels. Are you ever using the information of Qflag in the results or discussion? If it never comes up you might as well delete it.*

We will make a global legend, but believe it is good to be transparent about the data quality. We will add some discussion about the data quality in the main manuscript.

*Figure 6 the text on the x- and y-axis is too small*

We will increase the font size.

*Figure 8 It would be nice to stay consistent in the color choice for the models*

We will change the colors accordingly.

*Line 569 Aren't some of the models (all?) processed based and not empirical?*

We agree that our formulation here seems to indicate this, but we referred only to the more empirical components in these models, such as prescribed vegetation cover and rooting depths. We will try to remove this confusion in the revised manuscript.

*Figure S2.1, S2.4, S2.7, S2.10, S2.13, panel h – can you maybe add padding between 100% projected cover and the figure edge, the way it is now it looks like you're cutting off at 100%*

We will change this accordingly.

*In general, you have a lot of supplementary figures but don't refer to all of them in the manuscript.*

This is indeed true, we will add some more references to the supplementary material. We added relatively many, in order to provide full transparency and background information. Our supplementary information is also the output of the fully reproducible and re-usable modelling workflow we employed here, so it contains a bit more material than strictly needed for the paper. We will make this a bit clearer in the main paper and the introduction to the SI.

---

## Author Response (AR1)

**Final responses**

We would like to thank the editor and referees for their constructive feedback and assessment. We are especially happy that we could move forward with our suggested changes, and really tried to structure the manuscript concisely. Below, we first address the comments of the editor, followed by a list of changes related to comments of the referees and the editor. We added and updated our initial responses to the referees, with more details about the specific changes. In the following, editor and referee comments are written in italics.

**Response to Editor Comments**

*I have read carefully through the exchange and I agree that in the current stage misunderstandings may have contributed to the reviewers wish to change the narrative and analysis of the manuscript. I therefore advise to go forward with the proposed changes. Please pay special attention to the hypothesis. They should be well motivated from the introduction and their value understandable also for a reader with a general background in terrestrial biosphere models and not specifically with VOM. I also suggest reformulating them to be less technical and more accessible to a general audience. Please take care that sufficient model information is presented so that the core part of the manuscript can be understood without reference to another source. I strongly support your conceding to formulating the discussion more generally. Please make sure that the manuscript is structured concisely, and the structure is well-recognizable.*

Thank you for the careful assessment. We made substantial changes in the introduction, and hope we clarified our motivation for this research. We made additional changes to the hypotheses, and formulated them in a less technical way, as suggested. At the same time, we increased the model information in the methodology, including several key equations. We also added more general discussions and conclusions in the context of terrestrial biosphere modeling and optimality theory.

**List of changes**

- **Introduction**

The introduction was re-organized and largely reformulated. We tried to reduce the abundant discussion of individual studies, as mentioned by Referee #1, which often occurred in the part about model inter-comparisons. Hence, this was shortened and generalized (L.63-70). We added special paragraphs related to the specific challenges of optimality theory (L.85-126), which also link to the hypotheses, in order to address the concerns of the editor and Referee #2 and #3. We paid especially attention to the importance of the carbon costs for the plant hydraulic system (L.96-111), as this was generally mentioned by all referees, but also elaborated and discussed rooting depths including the references given by Referee #1. In addition, we discussed the importance of phenology (L.112-119), related to comments of Referee #2 and #3. Eventually, we re-organized the paragraphs to improve the flow, as asked for by Referee #3.

- **Hypotheses**

We reformulated the hypotheses in more general terms, based on the comments by the editor and Referee #2 and #3.

- **Study sites**

As suggested by Referee #1, we shortened the study site description, but added some extra details in Table 1 now. We also moved the table with the soil properties to Supplement S8, as suggested by Referee #1 and #3.

- **Vegetation Optimality Model**

We added more details to the model description as suggested by the referees and editor. First, the section about the water balance was extended, with also explanations about the water retention model, as requested by Referee #3. At the same time, we added the equations regarding the carbon costs and the net carbon profit (Eq. 1-4) to clearly define these, in order to address the comments of Referee #1 and #2. We separated the section about the vegetation optimization in long-term optimization and short-term optimization. In the section about the short-term optimization, we elaborated on how the VOM simulates phenology, as Referee #2 and #3 pointed out that this was previously lacking. At the same time, more details were given about the SILO-data in section 2.2.6, based on comments of Referee #1 and #3.

- **Modelling experiments and intercomparison**

We added more details about the models used by Whitley et al. (2016), related to the comments of Referee #2 and #3, and extended the section about the sensitivity to the water transport cost factor (Sect. 2.3.2) as suggested by Referee #1 and #2. We also added more details about the meaning of this cost factor here (L.293-296), as suggested by Referee #2.

- **Results**

We added an extra section describing the results when both rooting depths and vegetation cover are prescribed, as suggested by Referee #1. At the same time, we added some statements about a more structural model comparison (Supplement S7), mainly related to comments of Referee #1 and #3.

**- Discussion**

We added an extra section about the carbon costs for the water transport system to stress the importance of this (Sect. 4.3). As suggested by the editor and Referee #2, we also added paragraphs (end of Sect. 4.4 and 4.5) where we discuss more generally how the optimality theory, as applied here, compares to other approaches. Eventually, we decided to add a paragraph about data quality constraints, related to a comment of #Referee 3.

**- Conclusions**

We re-wrote the paragraph following the hypothesis of the carbon costs of the water transport system (L.623-628) to emphasize the importance of this. We also generalized our conclusions in the last part of this section, mainly related to the comments of Referee #2 and the editor.

**- Figures**

Figure 2 was changed according to the suggestions of Referee #3, with shaded areas for the dry and wet periods. We also removed the model runs of Schymanski et al. (2015), based on a comment of Referee #1. Following the other suggestions of Referee #3, Figure 3 was updated by removing the connecting lines and adding an arrow indicating the dryness, and we added a global legend and color bar in Figures 4, 7 and 9. We extended Figure 6 and added all different cases now for clarity.

**- Supplements**

Two extra supplements were added. Supplement S6 contains more detailed results about the model runs with prescribed vegetation cover and rooting depths, as proposed by Referee #1, whereas Supplement S7 contains a more systematic model comparison, related to remarks of Referee #1 and #3.

**Response to Referee #1**

We would like to thank Referee #1 for the review, which we see as very helpful. The Referee brings forward several valid points that we will improve on in a revised version of the manuscript. Below, we address the comments of Referee #1, with the referee comments written in italics.

*However, into details, I am not satisfied with the manuscript organization and the writing itself. Overall, I found there are many information currently included in the manuscript is unnecessary. The introduction is too long and contains many individual studies, which should be largely shorten with more highly-summarized findings/conclusion from existing individual studies. The detailed site description is also not needed, simply summarize the five sites with their specific properties listed in Table1. Table 2 is suggested to move to supplementary.*

We shortened the introduction, and condensed it, with less individual studies, and more general findings. More specific, we shortened the discussion of model inter-comparisons in the introduction, removing the abundant discussion of individual studies (L.63-70). Similarly, we also shortened the site descriptions and moved Table 2 to Supplement S8.

*On to content, I think the part of dealing with water transport cost parameter is more or less deviates from the main line. I would suggest remove the second hypothesis but describe how this parameter was chosen (either prescribed following previous studies or locally parameterized) in the manuscript. Then, the overall structure of the manuscript become: 1) test the VOM using site observations and compare it with TBMs; 2) what happened if remotely sensed vegetation cover was used? 3) what happened if prescribed rooting depth was used. Followed by discussion. I understand that the water transport cost parameter is also related to the overall performance of the model, but if that is included, why not other model parameters? And also you will need to describe you model in detail to allow readers who do not familiar with the model understand the role of this parameter in the model.*

The analysis of the water transport cost parameter was included as this is the only parameter in the model that was not based on literature values so far. It was originally tuned to achieve reasonable results at only one of the sites, Howard Springs, and hence we find it important to investigate in how far the same value of this parameter can be used at the other sites along the transect. This is crucial for assessing the utility of the optimality theory, which is the main goal of this study. We also explained the importance of this analysis more in the revised manuscript, and the necessity of this analysis for the interpretation of the other analyses. For that reason, we elaborated more about the carbon costs for the water transport system in the introduction (L.96-111), but also described it in more detail in the methods (Sect. 2.2.3 and 2.3.2).

The structure of the original manuscript was not much different from what the Referee suggested, but we followed the proposed structure more consciously in the revision: 1) test the VOM using site observations and compare it with TBMs; 2) find the underlying reasons for differences in model performances, with 1) what happens if we vary the unknown parameter for the water transport cost? 2) What happens if remote sensed vegetation cover was used? 3) What happens if prescribed rooting depth was used? As the water transport cost parameter is the most uncertain in the entire model, it is important to assess this parameter first, and then expand the analysis in order to find more underlying reasons for deviations from the observations.

Regarding the model details, we now refer the reader more clearly to the accompanying technical paper in GMD (https://doi.org/10.5194/gmd-2021-151, L215-216), as well as the previous papers with more

details about the VOM. We also expanded the section about the water balance (Sect. 2.2.1), long- and short term optimization (Sect. 2.2.4, 2.2.5) and added the equations for the carbon costs and the net carbon profit (Sect. 2.2.3).

*Another question is why not include a scenario that consider both prescribed vegetation cover and rooting depth, in comparison to the scenario with both vegetation properties optimized.*

This is a good idea, we conducted the suggested simulation and presented the details in the results Sect. 3.5 and Supplement S6.

*Other comments:*

*Line 215ï1⁄4infiltrate -> infiltration*

Changed accordingly (L.178).

*Line 216: Why 30m? Is this the defined soil depth in the model? Not sure if the choice of this depth impacts the modelling results.*

The 30m was chosen in order to represent freely draining conditions, with deep groundwater tables. See also the accompanying technical paper in GMD, where we found there is a strong influence. We also clarified this here (L.175-176).

*Line 225-230. This may present a source of uncertainty, as the observed fluxes are directly linked to the observed meteorological forcing at the sites, whereas the SILO data was used here to inform the model. Suggest to at least evaluate the used SILO data at each site against site-observed meteorological variables during their overlapping periods.*

Please see Supplement S4 of our accompanying technical paper in GMD (https://doi.org/10.5194/gmd-2021-151). We found that replacing the daily meteorological data from SILO, with aggregated daily data from the flux towers, did not lead to strong differences in the results. We clarified this here in the main manuscript as well (L.233-235).

*Line 223 Is there any published paper supporting this? Otherwise, simply states this information is measured at each site.*

We believe the referee means Line 232, where we will clarify the source of the soil data (now in L.237).

*Line 260-263 and the following sections. If I understood correctly, the last two hypotheses are related to replacing vegetation properties with prescribed values and the second hypothesis is about water transport cost parameter. Please check.*

We corrected this (L.265-268).

*Line 246 and throughout: evapotranspiration is often written without a hyphen.*

We are aware of this, but this is done on purpose. We try to emphasize here that it actually involves two different processes: evaporation and transpiration. See also our statements in L.255-256.

*Line 316-317: Where is the evidence for this? Figure 3. Please indicate where needed. In addition, in figure 2 at Howard Springs (but not for all other sites), there is a light green curve indicating the results of Schymanski 2015. What is the difference is model configuration between Schymanski 2015 and this study? And what is this for? It is not introduced.*

This can be seen in Figure 3b, we added a reference in the text (L.340).
We also removed the lines related to Schymanski et al. (2015), as this analysis is presented in the accompanying technical paper in GMD.

*Line 400. This is an overstatement. Looking at Figure 2, the VOM considerably overestimates GPP from observation and even compared with other TBMs.*

We rephrased this (L.438-439), following a more formal ranking of the models in Supplement S7, and added some sentences about this in the Results (L.325-327) and Discussion (L.444-446). The VOM achieved the top rank of the compared models in the combined performance of simulating ET and GPP, despite its weaknesses, especially in simulating GPP.

*Line 500. I do not agree with this hypothesis/statement. This may simply caused by the fact that the adopted VOM was not able to reproduce the actual rooting depth using the embedded optimality principles. Many previous studies have already demonstrated the importance of accurately representing rooting depth in the hydrological model to improve the modelled fluxes, for example, those by Kleidon and Heimann (1998) and more recently by Wang et al. (2016) and Yang et al. (2016).*

*Kleidon, A., and M. Heimann (1998), A method of determining rooting depth from a terrestrial biosphere model and its impacts on the global water and carbon cycle, Global Change Biol., 4(3), 275–286.*
*Wang-Erlandsson, L., W. G. M. Bastiaanssen, H. Gao, J. J€agermeyr, G. B. Senay, A. I. J. M. van Dijk, J. P. Guerschman, P. W. Keys, L. J. Gordon, and H. H. G. Savenije (2016), Global root zone storage capacity from satellite-based evaporation, Hydrol. Earth Syst. Sci., 20(4), 1459–1481.*
*Yang, Y., R. J. Donohue, T. R McVicar (2016), Global estimation of effective plant rooting depth: Implications for hydrological modeling. Water Resources Research, 52, 8260-8276.*

We fully agree that it is important to accurately represent rooting depths, but we formulated the alternative hypothesis in our paper originally, which we intended to reject. We rephrased the hypotheses now. We formulated this part also more clearly and discuss the suggested references in the introduction (L123-126) and the Discussion (L.558-560).

**Response to Referee #2**

We would like to thank Referee #2 for the review, and improved on the points brought forward by the referee. Below, we address the comments of Referee #2, with the referee comments written in italics.

*1. Hypothesis 1: One of my major concern concern it that the TBMs (or LSMs) in Whitley et al. (2016), which are designed for global-scale, long-term climate projection models, are optimized to make a fair comparison to VOM, though they have more specific vegetation parameters.*

We did not re-run any of the TBM simulations of Whitley et al. (2006) in order to compare to the VOM, we re-used the model results of Whitley et al. (2016), where the TBMs were applied independently of the VOM. Of course, the modellers involved in the study of Whitley et al. (2016) applied different levels of site-specific parametrisation, and we do not know in how far any parameters were optimized in the original study. With this hypothesis, we mainly try to assess where our optimality-based model stands, and if it still gives satisfactory results in comparison with models that were applied in what the specific modellers considered the "best" way possible. We added now more details about the other models in Sect. 2.3.1 Model intercomparison.

*2. Hypothesis 2 is too model specific. Does any other TEMs use this paramter? or what would other models learn from this? or does this hypothesis have any implication for plant adaptation or optimality?*

Plant hydraulics is currently being implemented in TBMs as a major limitation for water use during drought, whereas in the VOM, it is only represented in the form of the water transport cost factor. The intention of this hypothesis is to test whether a general water cost factor across sites leads to reasonable results or if site-specific cost factors would yield significantly improved results. Therefore, assessing this cost factor is also highly necessary in order to assess the general concept of the optimality theory, as applied here. We clarified this in the revised version in the Introduction (L.96-111), but also reformulated this hypothesis in more general terms.

*3. Hypothesis 3 is also model specific. It depends on model representations of the links between LAI, vegetation cover, light transfer and absorption, assimilated carbon allocation. This manuscript does not clear describe how these processes are represented in VOM.*

We clarified in the revised manuscript that the prognostic simulation of phenology (LAI and vegetation cover dynamics) is a central concern of vegetation models and optimality theory (L.112-119). We also assess here how the optimality theory, as applied in the VOM, can be improved, and therefore we should also assess the prognostic simulation of the projective cover. As requested, we will explain in more detail how phenology is simulated in the VOM and hence provide a better context for this hypothesis. We did this by adding a specific section about the short-term optimization (Sect. 2.2.4) in the Methods-section. At the same time, we re-formulated the hypothesis in more general terms. Thank you for pointing out this gap.

*4. Hypothesis 4 should be the focus of this manuscript. If this would have been well tested, it is enough to be a good paper. However, this would be greatly affected by the representation of subsurface soil moisture profile and further linked to capillary fringe. I am wondering why the authors did not use the results of Schymanski et al. (2015). So I strongly suggest to include groundwater effects.*

We agree with the referee that groundwater effects are important, but none of the TBMs treated in Whitley et al. (2016), considered them, so we did not in this paper, either. In the revised manuscript, we point the reader to the accompanying technical note for a systematic comparison with the results of Schymanski et al. (2015) in terms of groundwater influence (https://doi.org/10.5194/gmd-2021-151).

*5. It is not clear to me (I have to read also Schymanski et al., (2015)) that how VOM is optimized. "Maximizing the NCP"? what is the maximum NCP? how do we know the maximum NCP? Please expand Section 2.2.4 a bit to describe this process in model detail.*

Thank you for pointing out this lack of clarity. We added more details in this section. More specifically, the SCE-algorithm samples the long-term vegetation properties. With this, the VOM is run, and over the full period, the total $CO_2$-assimilation minus the total carbon costs represents the Net Carbon Profit. The vegetation properties that achieve the highest NCP within the parameter space are kept and considered as the optimal vegetation properties. We added these details now in Sect. 2.2.4 and 2.2.5, and also added the equations that define the NCP and the carbon costs (Equations 1-4).

*6. Section 2.3.3, the cost factor for water transport (crv) should be described here. If not reading Schymanski et al., (2008), I did not understand its meaning.*

We added a specific paragraph describing the cost factor for water transport (L.293-296), but also added the equation for the carbon costs (Eq. 3). In the Introduction, L96-111, we introduced more clearly the challenges related to this cost factor, but also explained more in Sect. 2.3.2.

*7. Conclusions: please generalize these conclussions through discussions. I do not care much about how VOM is better or not or how to improve it but more about how to implement improved understandings through VOM studies into the current TBMS that arewidely used in IPCC climate projections.*

Point taken. We re-wrote the conclusions from the perspective of general benefits of optimality modelling, and added more general implications of our findings (especially in L.672-684). We emphasized the identified deficiencies and possible improvements of the VOM, and related them to the general understanding of the applied optimality theory. At the same time, we added more discussion at the end of Sect. 4.4 and 4.5, where we compared the optimality-based approach with other approaches.

**Response to Referee #3**

We would like to thank Referee #3 for the review, and will improve on the points raised by the referee. Below, we address the comments of Referee #3, with the referee comments written in italics.

*In general, I felt the authors could have been a bit more careful with their writing. For example, they confuse their hypotheses throughout the manuscript (see below specific comments) which sometimes makes it hard to follow their argumentation. The introduction is a bit too long and, in parts, not very well linked. I liked how the authors explicitly point out the four hypotheses they aim to explore, however, it is not quite clear to me how hypotheses 2 and 3 emerge from the introduction. It further would be nice to have more details on the model description – you could move table 2 to the supplement for example to make more room. Your discussion is detailed but I'd like to see more about possible future directions.*

We are sorry that the numbering of hypothesis was confused a few times and thank the reviewer for picking this up. We carefully checked this in the revised manuscript. We also improved the introduction by shortening it and improving the flow, as requested.
We tried to introduce the motivation for Hypotheses 2 and 3 in the introduction initially, but made this more explicit in the revised manuscript (L.96-119). Briefly, plant hydraulics are currently being implemented in TBMs as a major limitation for water use during drought, whereas in the VOM, it is only represented in the form of the water transport cost factor. For this reason, we test with this hypothesis whether a general water cost factor across sites leads to reasonable results or if site-specific cost factors would yield significantly improved results.
Regarding Hypothesis 3, we emphasized more in the introduction that the prognostic simulation of phenology (LAI and vegetation cover dynamics) is a central concern of vegetation models and optimality theory (L.112-119). For this reason, a systematic evaluation of the prognostic vegetation cover in comparison with more conventional approaches (i.e. prescribed values), should show how the optimality theory, as applied in the VOM, extends the capability of conventional models and how it can be further improved. We also discussed more explicitly future directions in this context, as requested.
We included a detailed model description in the accompanying manuscript in GMD (https://doi.org/10.5194/gmd-2021-151), which is why there are not too many details in this manuscript. However, also based on the comments of the other referees, we added more information now (additional equations, extended description of the water balance model and more details about the short- and long-term optimization) and referred more explicitly to this technical note for details. We also moved Table 2 to Supplement S8, as suggested.

*Specific comments*

*Line 41-44 The sentence is very long and hard to follow*

We rephrased and shortened this sentence (L.66-68).

*Line 61 Doesn't the default version of LPJ-GUESS have more than five plant functional types?*

Thank you for pointing this out, we eventually removed this statement.

*Line 91-92 The contents of the sentence are not linked very well*

We removed this sentence eventually.

*Line 98 Increase or decrease in annual rainfall?*

We added here that it is a decrease towards the south (L.48).

*Line 118 'optimizing vegetation properties to maximize the NCP'?*

Changed accordingly (L.82-83).

*Line 136 Do timescales of precipitation matter? I.e. is annual PPT driving the rooting depth or are seasonal timescales more important?*

We added more discussion about the rooting depths (L.120-126). In the VOM, rooting depths are a result of the long-term optimization of the roots, and do not only depend on climate but also on hydrology, i.e. the water storage capacity of the soil and the distance to groundwater.

*Line 136 Therefore instead of but? '[...] therefore is likely to change over […]'*

We changed this to "and are, therefore, likely to change over…"(L.122-123).

*Line 162 In table 1 it's AU-How*

We shortened this section and removed this reference, but corrected this throughout the manuscript.

*Line 173 In table 1 it's AU-DaS*

We removed this reference in the text due to the shortening of the section.

*Line 199 '[...] is defined by maintenance respiration, projected cover to the turnover and maintenance of leaf area' – I find this sentence a bit unclear*

This sentence was eventually removed.

*Line 208 How can seasonal vegetation cover vary on a daily basis? Maybe rephrase*

We clarified that in the VOM, seasonal vegetation cover is allowed to vary slightly from day to day, resulting in a seasonality with a maximum during the wet season, and a minimum during the dry season. We elaborated on this in Sect. 2.2.4 Short-term optimization.

*Line 225-230 Does SILO provide point data or are the site met data derived from a spatial dataset (if yes which resolution?) I understand the argument that a longer timeseries helps to run the model, but it would be nice to see any sort of comparison between the observed met data at the site and the SILO dataset. I guess in general I would just like to have more information about the input forcing to get an idea about the uncertainty. Do the models from the Whitley paper run with the same meteorological forcing or do they use the data collected at the site?*

We added more details about the SILO-data (L.226-227). We also referred to our accompanying technical paper in GMD (https://doi.org/10.5194/gmd-2021-151, Supplement S4), where we replaced the daily meteorological data from SILO with aggregated daily data from the flux towers. Eventually, this did not lead to strong differences in the results. We added statements about this in L.231-235. The models in Whitley et al. (2016) were generally run with the flux tower data, except for BIOS2 that also used a gridded product. We added this in the text as well (L231-233, L.276-279).

*Line 235-236 Can you provide a bit more detail about the water retention model? It was never mentioned before*

We added more details about the water retention model (L.170-174). There are also more model details in the accompanying technical note in GMD ([https://doi.org/10.5194/gmd-2021-151](https://doi.org/10.5194/gmd-2021-151)).

*Line 247 Can you specify what the 'usual energy fluxes' are*

We referred here to observations of incoming and reflected solar radiation. We clarified this (L.251-253).

*Line 249 LE has already been introduced two sentences earlier*

We corrected this (L.254).

*Line 264 Isn't the last hypothesis about rooting depth?*

We are sorry for the confusion, we changed the order in a previous version of the manuscript. We carefully checked this and corrected it here (L.265-268) and throughout the manuscript. Thank you for pointing this out.

*Line 265-276 I might have just overlooked in your submission – but can you describe in more detail what the model set up is for the model intercomparison you use from the Whitley et al paper? Surely there will be more detail in the Whitley paper to help understand but while reading your submission I was for example wondering whether there are changes to some of the parameters in the models to capture the site specifics better or whether they 'just' ran in their original configuration with the meteorological forcing from the sites […]*

We added more specifics about the models in the revised manuscript (L.276-284).

*Line 277 Third and fourth hypotheses?*

Thank you, we corrected this (L.306).

*Line 294 In the introduction it says -0.1-0.1 for the cost factor for water transport (but I might have understood?)*

In the introduction, we discussed the variation, instead of the absolute value. We removed this from the introduction for clarity.

*Line 297 second hypothesis (also Line 298)*

Changed accordingly (L.300).

*Line 300 'Regardless of the result here' can you explain why you make this decision?*

We used this value, as it was the outcome of the sensitivity analysis by Schymanski et al. (2015). The assumption was that this cost parameter is valid with this value for all sites, while here we assess the original assumption using a sensitivity analysis. We clarified this more in the revised manuscript (L.295-296, and in the Introduction L.98-103).

*Figure 2: Maybe include shaded areas for dry and wet season, but also include dry and wet season months in caption. Can you include what the ensemble years are too? Panel e says Daly Uncleared but it was referred to as Daly River before*

Thank you for these suggestions, we made corrections accordingly in Figure 2. Daly River and Daly Uncleared refer to the same site, we made this more consistent throughout the manuscript and the figures.

*Line 306 Can you define dry/ wet season (which months)?*

The wet season is from December-March, and the dry season from June-September. We added this (L.329, L.331, captions Figure 2 and 6).

*Line 327 Not sure, it looks like the minimum is quite similar for the models but for the maximum values, LPJ-GUESS and MAESPA seem to be too low*

Thank you for pointing this out, we formulated this sentence more carefully (L.349-352).

*Line 400 I'm not sure I agree with this. It may be true when you look at the annual values in figure 3 but based on figure 2 you can't really reach this conclusion.*

We rephrased this (L.438-439), our main point is that the VOM does not perform substantially worse than the other models. We added also a more systematic model comparison in Supplement S7 and statements about these findings in L.325-327 and L.443-446.

*Figure 3: Why are the data points connected in panel a and b? It already is hard to distinguish the data points, the lines make it even harder. You also do not connect them in the other panels – it might be nicer to be consistent. Further, it might be helpful to offset the points in a and b (like in c-f). You could also increase the size of the observation marker, it gets lost in all the other points. Lastly, it could be helpful to include an arrow below the lower x-axes indicating whether the sites go from dry to wet or the other way around (but a lot of this is personal reference of course)*

We tried to assess the patterns over the transect, which is why we added the line. But we agree that it is clearer without the lines, so we re-plotted Figure 3 without lines. We also like the other suggestions, and added an arrow indicating the dryness and increased the size of the observation markers. Thank you for the suggestions.

*Figure 4 and 7: Maybe use a global legend and remove obs legend from panels. Are you ever using the information of Qflag in the results or discussion? If it never comes up you*

*might as well delete it.*

We made a global legend for Figures 4,7 and 9, but believe it is good to be transparent about the data quality. We added some discussion about the data quality in the main manuscript (Sect. 4.6).

*Figure 6 the text on the x- and y-axis is too small*

We increase the font size in Figure 6.

*Figure 8 It would be nice to stay consistent in the color choice for the models*

We changed the colors in Figure 8 accordingly.

*Line 569 Aren't some of the models (all?) processed based and not empirical?*

We agree that our formulation here seems to indicate this, but we referred only to the more empirical components in these models, such as prescribed vegetation cover and rooting depths. We rephrased this to remove this confusion in the revised manuscript (L.673).

*Figure S2.1, S2.4, S2.7, S2.10, S2.13, panel h – can you maybe add padding between 100% projected cover and the figure edge, the way it is now it looks like you're cutting off at 100%*

We changed this accordingly.

*In general, you have a lot of supplementary figures but don't refer to all of them in the manuscript.*

This is indeed true, we added some more references to the supplementary material. We added relatively many, in order to provide full transparency and background information. Our supplementary information is also the output of the fully reproducible and re-usable modelling workflow we employed here, so it contains a bit more material than strictly needed for the paper. We also pointed this out in the main paper (L.148-149) and the introduction to the SI.

---

## Author Response (AR2)

Dear editor,

We are very happy with the decision to accept our manuscript. We just made the following technical corrections:

- Correction of the caption of Figure 2.
- Correction of NOAH-MP to Noah-MP.
- Addition of the zenodo link to the repository with our code and data, in the text as a footnote, and in the *Code and data availability* section.
- Addition of the reviewers to the *Acknowledgments*.

And of course, we wish you a merry Christmas as well, with all the best for the new year.

On behalf of all authors,
Remko Nijzink